# Theoretical Modeling of Chemical Equilibrium in Weak Polyelectrolyte Layers on Curved Nanosystems

**DOI:** 10.3390/polym12102282

**Published:** 2020-10-05

**Authors:** Estefania Gonzalez Solveyra, Rikkert J. Nap, Kai Huang, Igal Szleifer

**Affiliations:** 1Department of Biomedical Engineering, Northwestern University, Evanston, IL 60208, USA; estefania.solveyra@northwestern.edu (E.G.S.); rnap@northwestern.edu (R.J.N.); huangk05@gmail.com (K.H.); 2Chemistry of Life Processes Institute, Northwestern University, Evanston, IL 60208, USA; 3Department of Chemistry, Northwestern University, Evanston, IL 60208, USA

**Keywords:** chemical equilibrium, pH-related processes, end-tethered weak polyelectrolytes, theoretical methods, charge regulation

## Abstract

Surface functionalization with end-tethered weak polyelectrolytes (PE) is a versatile way to modify and control surface properties, given their ability to alter their degree of charge depending on external cues like pH and salt concentration. Weak PEs find usage in a wide range of applications, from colloidal stabilization, lubrication, adhesion, wetting to biomedical applications such as drug delivery and theranostics applications. They are also ubiquitous in many biological systems. Here, we present an overview of some of the main theoretical methods that we consider key in the field of weak PE at interfaces. Several applications involving engineered nanoparticles, synthetic and biological nanopores, as well as biological macromolecules are discussed to illustrate the salient features of systems involving weak PE near an interface or under (nano)confinement. The key feature is that by confining weak PEs near an interface the degree of charge is different from what would be expected in solution. This is the result of the strong coupling between structural organization of weak PE and its chemical state. The responsiveness of engineered and biological nanomaterials comprising weak PE combined with an adequate level of modeling can provide the keys to a rational design of smart nanosystems.

## 1. Introduction

The use of end-tethered polymers and polyelectrolytes has been ubiquitous in a wide range of applications. As they allow modulating and influencing the surface properties, they have been used in surface modification for colloidal stabilization, lubrication, adhesion, or wetting of the surface [1,2,3]. More recently they have also become the focus of theoretical and experimental studies for potential nano-technological and biomedical applications [4]. For example, polymer-modified nanopores show promise in applications such as water purification [5], nanofluidic circuits [6], and chemical sensing [7]. Likewise, polymer-tethered nanoparticles and micelles have potential biomedical applications including drug delivery devices, contrast agents, or as platforms to deliver targeted therapeutics [8,9], while biocompatible polymer brushes can be used to prevent nonspecific adsorption of proteins and cells [10].

The structure of grafted layers composed of neutral polymer are determined and controlled by physical interactions such as van der Waals forces (solvent quality), excluded volume, and hydrogen bonding. When dealing with polyelectrolytes (PEs), electrostatic interactions come into scene playing a decisive role. One can distinguish between two classes of PE brushes: strong PE-brushes in which monomers within the chains are permanently charged (also referred to in the literature as quenched PE-brush) and weak PE-brushes, in which monomers exhibit acid–base behavior (also described as annealed PE-brushes). In both cases, electrostatic interactions between charged segments can be tuned by changing the ionic strength of the bulk solution. Their behavior with respect to the bulk pH is what distinguishes them apart: while the amount of charge in strong PE-brushes is fixed and independent of bulk solution pH, this becomes a crucial variable for weak PE-brushes. Weak PE can be either acids or bases, with their degree of charge being controlled by the chemical reaction of (de)protonation, as we will discuss in Section 2. In this way, the structure and properties of weak PE layers can be controlled, beside conventional ways of solvent quality and temperature, also by pH and ionic concentration, granting them pH-sensitive features. This opens the possibility to designing functional materials that can change their properties by external cues like pH [4].

pH-related processes are crucial in biological and technological systems. They provide the mechanism to engineer smart nanoplatforms that are environmentally responsive and have maximized specificity. The basic idea is to take advantage of the acid–base properties of weak PE and the way their properties change when they are immobilized onto surfaces of different curvature and subject to different environmental conditions. Mechanisms of pH-responsiveness include pH-dependent swelling and dissolution, aggregation, dissociation, pH-labile linkers, pH-sensitive drug-polymers conjugates, etc. [11]. In all of them, the system tries optimizing the physical and chemical interactions by coupling molecular organization and charge state in a complex manner. We will address the mechanism of charge regulation in Section 3 and provide several examples thereof in Section 4.

The molecular details and insights provided by adequate modeling and theoretical frameworks are key to rationalizing experimental observations and for allowing a more complete description of the processes taking place in the system. Neutral polymer layers have been the subject of many theoretical and experimental studies, which resulted in a fairly complete description of their behavior [12]. Strong polyelectrolyte layers have also been the subject of a considerable amount of research [13]. Meanwhile, theoretical investigations into the behavior of weak polyelectrolyte layers is more recent and its theoretical understanding is less complete. Invariably, these methods use previous theoretical approaches developed to investigate neutral polymers and strong PE layers as a starting point. They include theoretical approaches such as scaling theory, analytical and numerical self-consistent field theory, and density functional theory. Consequently, these approaches inherit the advantages and weak points of those neutral polymer theories and add to them the complexity of electrostatic and chemical interactions.

In this context, we want to present an overview of some of the theoretical methods that we consider key in the field of weak PE at interfaces, to enable a better understanding of the various theoretical approaches. It is worth mentioning that our presentation in this short review is far from complete, so that the reader is strongly encouraged to explore the list of references for further details and other methods omitted in the text. We have selected some main theoretical methods that we believe are key examples for an overview of the past, present, and future of this field. Our intention is not to present an exhaustive review of the available theoretical literature devoted to weak polyelectrolyte layers, but to focus on the common features and differences of several of the most used theoretical methods, presenting examples as well as key findings and insights that have emerged from the study of weak polyelectrolyte layers, particularly when grafted to curved nanosystems, like nanoparticles and nanopores.

This review article is organized as follows. In Section 2, we present a brief review of the thermodynamic description of chemical equilibrium, focusing on acid–base equilibrium. There, we present a common set of basic thermodynamics concepts that will be then followed by Section 3 with chemical reactions that can be included in various theoretical descriptions or frameworks of polymer layers. Given the authors’ background in thermodynamic-statistical free energy approaches, the primary focus will be on these methods (although not exclusively), including self-consistent field theory and lattice theory, like the Scheutjens–Fleer theory. Simulations methods will also be mentioned, although not discussed in detail. In Section 4, we provide examples of applications chosen to illustrate some key insights that have emerged from the theoretical approaches discussed. The reviewed literature refers mainly to weak PE end-grafted to surfaces of different curvatures, such as nanoparticles, nanopores, and nanochannels. We propose going a little bit beyond acid–base equilibrium, presenting a couple of other chemical equilibria that can be treated similarly, and that we find very relevant for technological and biological applications. We finish this review by a general conclusion and perspectives for the future work in the field in Section 5.

## 2. Chemical Equilibrium

Thermodynamically, the chemical equilibrium of a weak acid, such as carboxylic acid,
(1)$$\mathrm{AH}⇌A^{-}+H^{+},$$
is determined by the chemical equilibrium constant, which is the product of the activities of the products divided by the activity of the reagent.
(2)$$K_{a}^{⦵}=\frac{a_{A^{-}}a_{H+}}{a_{AH}}\approx \frac{\left[A^{-}\right]\left[H^{+}\right]}{[AH]c^{⦵}}=\frac{K_{a}}{c^{⦵}}.$$

Here, $K_{a}^{⦵}=exp\left(-\beta \Delta G_{a}^{⦵}\right)$ is the chemical equilibrium constant, which is related to the standard Gibbs reaction free energy $\Delta G_{a}^{⦵}=\mu_{A^{-}}^{⦵}+\mu_{H^{+}}^{⦵}-\mu_{AH}^{⦵}$. In dilute solution and for simple acids, one can approximate the activities of the molecules by their concentrations and end up with the familiar form for the chemical constant $K_{a}=\left[A^{-}\right]\left[H^{+}\right]/\left[AH\right]$, where $K_{a}$ corresponds to the experimental reported equilibrium constant and *c*^⦵^ corresponds to the reference state solution. We explicitly retained *c*^⦵^ in the above equation to emphasize that $K_{a}^{⦵}$, unlike $K_{a}$, is dimensionless. Conventionally the reference state corresponds to a 1M solution. Therefore, $K_{a}$ has units of molarity.

Given the equilibrium constant, one can determine the extend of the reaction, *f*, or the fraction of acids that are deprotonated. Namely,
(3)$$f=\frac{\left[A^{-}\right]}{\left[A^{-}\right]+\left[AH\right]}=\frac{1}{1+\left[H^{+}\right]/K_{a}}=\frac{1}{1+10^{pK_{a}-pH}}.$$

Explicitly the pH is defined as the $pH=-\log_{10}\left[H^{+}\right]$. The $pK_{a}$ is defined in a similar fashion. The above textbook equation tells us that the fraction of charged acids in solution is determined solely by the pH of the reservoir and the $pK_{a}$, which is an intrinsic property of the acid. The degree of charge is independent of the amount of acid in solution, reflecting the fact that Equation (Equation 3) is only valid for a dilute acid in solution. This was also explicitly implied by the approximation of the activities by concentrations.

However weak polyelectrolytes, even in dilute solution, do not obey ideal-solution behavior as given by Equation (Equation 3) [14]. The connectivity of the polyelectrolyte chain results in electrostatic repulsions between the charged monomers. Consequently, the degree of charge of a polyacid in solution is also influenced by the monomer concentration, molecular weight, salt concentration, etc. This was already realized by Katchalsky and coworkers in the 1940s in their pioneering work on weak polyelectrolyte solutions [14,15,16,17].

Now for the end-tethered polyelectrolytes, not only do the monomers behave non-ideally, but they are also not in a dilute environment. As a polyelectrolyte layer consists of chain molecules that are end-grafted relatively closely together to a surface, the monomers are in a concentrated environment. If the monomers are also charged, they experience large mutual electrostatic repulsions as well, given the high concentration of monomers in the brush. Therefore, we can not use $K_{a}=\frac{\left[A^{-}\right]\left[H^{+}\right]}{\left[AH\right]}$ any more. We need to consider and compute the activities coefficients of the monomers. Moreover, the interface makes the system anisotropic and the polymers are inhomogeneously distributed, thus we also need to replace the notion of uniform degree of dissociation and take into account that the degree of dissociation is position-dependent.

## 3. Theoretical Methods

Most theories, except simulations, involve expressing the free energy of the system as a functional of the density of all involved species, i.e., $F[\rho_{i}\left(\vec{r}\right)]$. This free energy functional contains contributions describing short-range interactions, long-range interactions, and entropic contributions. For weak polyelectrolytes, an additional contribution needs to be included to describe the acid–base equilibrium.

The free energy of end-tethered weak polyelectrolytes can be summarized as follows [18,19],
(4)$$F=-TS_{\mathrm{mix}}+E_{\mathrm{elect}}+F_{\mathrm{chem}}+E_{\mathrm{rep}}+F_{\mathrm{conf}}$$

The first term in the free energy, $-TS_{\mathrm{mix}}$, describes the mixing or translational entropy of the solvent (water) and mobile ionic species in solution and is given by the expression $-\frac{S_{mix}}{k_{B}}=\sum_{i}\int d\vec{r}\rho_{k}\left(\vec{r}\right)\left(\ln\rho_{i}\left(\vec{r}\right)v_{w}-1\right)$. The variable $\rho_{i}\left(\vec{r}\right)$ is the number density of mobile species $i=w,Na^{+},Cl^{-},H^{+},OH^{-}$ and $v_{w}$ is the volume of a water molecule, which is used as the unit of volume. The second term in Equation (Equation 4), $F_{\mathrm{elect}}$, stems from the electrostatic contribution to the free energy. Explicit functionals can be found in, e.g., [20,21]. Importantly, it is a functional of the electrostatic potential $\psi (\vec{r})$ and the total charge density $\langle \rho_{q}\left(\vec{r}\right)\rangle$, which is the sum of the charge density of all charged mobile ions and the charged polyelectrolytes, $\langle \rho_{q}\left(\vec{r}\right)\rangle =\sum_{i}ez_{i}\rho_{i}\left(\vec{r}\right)+\left(-e\right)f\left(r\right)\langle \rho_{p}\left(\vec{r}\right)\rangle$. Here, $z_{i}$ is the valance of the charged mobiles species and $\langle \rho_{p}\left(\vec{r}\right)\rangle$ is the density of the chargeable monomers at position $\vec{r}$, where $f(\vec{r})$ is equal to the position-dependent fraction of polyacid monomers that are actually deprotonated. Conversely, $\left(1-f\left(\vec{r}\right)\right)$ corresponds to the fraction of polyacid monomers that are protonated at position $\vec{r}$. Variation of $E_{\mathrm{elect}}$ with respect to the electrostatic potential results in the Poisson Equation [20,21].

The third free energy contribution, $F_{\mathrm{chem}}$, describes the acid–base chemical equilibrium and the chemical free energy associated with the (de)protonation of the polyacid monomers [22].
(5)$$\beta F_{\mathrm{chem}}=\int d\vec{r}\langle \rho_{p}\left(\vec{r}\right)\rangle \left[f\left(\vec{r}\right)\left(\ln f\left(\vec{r}\right)+\beta \mu_{A^{-}}^{⦵}\right)+\left(1-f\left(\vec{r}\right)\right)\left(\ln\left(1-f\left(\vec{r}\right)\right)+\beta \mu_{AH}^{⦵}\right)\right].$$

The first and third terms of Equation (Equation 5) describe the mixing entropy of the deprotonated charged state ($A^{-}$) and protonated state (AH), respectively. The second and fourth terms in Equation (Equation 5) correspond to the standard chemical potential of the charged and uncharged states. These are the same chemical potentials as the ones appearing in the reaction Gibbs free energy.

The fourth contribution to the free energy in Equation (Equation 4), $E_{\mathrm{rep}}$, describes the repulsive excluded volume interactions between all molecules, which are accounted for by assuming that the system is incompressible at every position $\langle \phi_{p}\left(\vec{r}\right)\rangle +\sum_{i}\phi_{i}\left(\vec{r}\right)=1.$ Here, $\phi_{k}\left(\vec{r}\right)=\rho_{k}\left(\vec{r}\right)v_{k}$ is the volume fraction of mobile species *k* with $v_{k}$ corresponding to its volume and $\langle \phi_{p}\left(\vec{r}\right)\rangle$ corresponds to the polymer volume fraction. These volume packing constraints are enforced through the introduction of Lagrange multipliers, denoted as $\pi (\vec{r})$. (Additional non-electrostatic van der Waals interactions, described the solvent quality, can potentially be included as well. See, for example, in [19].)

Finally, one needs to describe the polyelectrolyte layer, i.e., account for $\langle \phi_{p}\left(\vec{r}\right)\rangle$. A thermodynamic description of polyelectrolytes requires that the chain connectivity and the conformational statistics are taken into account. This conformational free energy contribution is denoted as $F_{\mathrm{conf}}$. In the next section, we shall discuss several different theories that describe $F_{\mathrm{conf}}$.

Assuming that the conformational free energy $F_{\mathrm{conf}}$ does not explicitly depend on $f(\vec{r})$, the fraction of protonated acid monomers is only coupled in the free energy with the polyelectrolyte density via $F_{\mathrm{chem}}$, then minimization of the free energy results in following reaction equilibrium equation.
(6)$$\frac{f(\vec{r})}{1-f(\vec{r})}=K_{a}^{⦵}\frac{e^{-\beta \pi \left(\vec{r}\right)\Delta v_{a}}}{\rho_{H^{+}}\left(\vec{r}\right)v_{w}}\;⟺\;\frac{\left[A^{-}\left(\vec{r}\right)\right]\left[H^{+}\left(\vec{r}\right)\right]}{\left[AH\left(\vec{r}\right)\right]}=K_{a}e^{-\beta \pi \left(\vec{r}\right)\Delta v_{a}}$$

Here, $\Delta v_{a}=v_{A^{-}}+v_{H^{+}}-v_{AH}∼v_{H^{+}}=v_{w}$ is the change in volume between reactant and products. The expression on the left for the position dependent degree of protonation can be rearranged to yield the equation shown on the right. Comparison of that equation with the equation for the degree of deprotonation of a monomeric acid (Equation 3), shows that $K_{a}^{⦵}$ is the true equilibrium constant controlling the chemical equilibrium and not the bulk chemical equilibrium constant $K_{a}$.

Similar to the equation for the bulk equilibrium constant, Equation (Equation 3), one can define a position dependent equilibrium constant $K_{a}\left(\vec{r}\right)=K_{a}\exp\left(-\beta \pi \left(\vec{r}\right)\right)\Delta v_{a}$. The position-dependence of this effective equilibrium constant emphasizes the fact that changes in the local environment influence the local degree of deprotonation, which is directly changed by the position dependent $\pi (\vec{r})$. These $\pi (\vec{r})$’s are the Lagrange multipliers or local osmotic pressure enforcing the incompressibilty constraints [19]. The degree of deprotonation is also indirectly changed, as it is determined by the local osmotic pressure and electrostatic potential, which in turn are coupled with the local volume and charged densities of the polyelectrolyte, solvent, and ionic species. Thus, the chemical equilibrium shows that the degree of local deprotonation can be very different from that of the ideal solution behavior because of the electrostatic potential ($\psi (\vec{r})$) as well as the local osmotic interaction ($\pi (\vec{r})$) can substantially deviate from their bulk values.

It should be apparent from the coupling of local deprotonation with the electrostatic potential and the local osmotic interactions that precise values of the polymer density, which results form the conformational free energy, are very important. In the next section, we shall discuss several mean-fields that describe $F_{\mathrm{conf}}$. The major difference between them is their treatment of chain connectivity and the polymer conformational entropy. They differ on the level of molecular details in which the polymer chains are described, ranging from very detailed including many molecular features, to a coarse, zoomed out description of the chain molecules.

### 3.1. Molecular Theory

In this approach, a large set of different polymer conformations is input into the theory and the free-energy minimization determines the probability of each of these polymer conformations [19,23]. Explicitly, the conformational entropy of a end-tethered polymer is given by [19,23]
(7)$$-\frac{S_{conf}}{k_{B}}=\sum_{g}\sum_{\alpha_{g}}P\left(\alpha_{g}\right)\ln P\left(\alpha_{g}\right),$$

Here, $P(\alpha_{g})$ is the probability of finding an end-tethered polymer chain in the conformation $\alpha_{g}$ that is end-tethered to a grafting point *g*.

In principle, the sum in the conformational entropy needs to be extended over all possible conformations. Only for small molecules like lipids and surfactant molecules one can exactly enumerate all conformations. For longer polymer chains the number of conformations becomes very large. Instead Monte Carlo or Molecular Dynamics (MD) simulations are used to generate a large representative set of self-avoiding polymer conformations. In this way intramolecular steric repulsions of the chain are taken into account exactly, while the intermolecular excluded volume interactions are treated in a mean-field approximation by assuming the system is incompressible. For tethered polymers or polymers near an interface, the MT approach works in general very well as corroborated by the good agreement between predictions of MT and experimental observations as well as computer simulations [24,25,26,27,28,29].

The probability distribution or pdf is the central quantity with the MT framework, because given the pdf, one can compute any structural and thermodynamic quantity related to the polymers, for example the polymer volume fraction. Explicit expressions for the pdf can be found, e.g., in [19,30]. Here, it suffices to observe that the pdf is a function of the osmotic pressure ($\pi (\vec{r})$) and electrostatic potential, degree of charge ($f(\vec{r})$), as well as the explicit position and type of all monomers, i.e., the conformation $\alpha_{g}$. The pdf determines the polymer density which is coupled with the local osmotic pressure and electrostatic potential. Thus, the degree of deprotonation Equation (Equation 6) is also coupled in a nontrivial and nonlinear fashion to the pdf, which in turn affects the local osmotic pressure and local electrostatic potential. This coupling between polymer density and local degree of deprotonation will also be discussed in the Applications section [19,30].

### 3.2. Self-Consistent Field Approach

In the Self-Consistent Field Theory (SCFT or SCF) approach the polymer chain is not represented explicitly but is instead viewed as a continuous Gaussian chain and a conformation of the polymer chain is represented by a continuous space curve ${\vec{r}}_{g}\left(s\right)$, where *s* is a contour variable that indexes the location of the segment and *g* denotes a tethered point. Now, making use of the properties of the Gaussian distribution, one can derive the so-called Edwards diffusion equation:(8)$$\frac{\partial }{\partial s}G_{g}\left(\vec{r},s\right)=\frac{b^{2}}{6}\nabla^{2}G_{g}\left(\vec{r},s\right)-\omega_{p}\left(\vec{r}\right)G_{g}\left(\vec{r},s\right).$$

Here, $G_{g}\left(\vec{r},s\right)$ is the segment distribution function that describes the probability of finding a polymer chain segment of length *s*, located as position $\vec{r}$ whose chain is anchored to the surface at position $\vec{r_{g}}$. The Edwards equation is the central equation in the SCFT approach, as given the segment distribution the polymer density can be computed. Here, $\omega_{p}\left(\vec{r}\right)$ can be viewed as the segment potential whose role is to represent the excluded-volume interactions as well as account for the solvent quality and electrostatic interactions.

Technical details and derivation of the SCFT equations can be found, for example, in [31,32,33,34,35,36,37,38]. Since its original formulation [39], it has been extended and improved and applied to many different polymer systems [32,33,34], including neutral tethered polymers [33,34,35,36,37,38,40,41] and tethered weak polyelectrolytes [42,43,44,45], as well as weak polyelectrolytes melts and in solution [46].

In the SCFT, the application of the segment distribution to compute the density, implies that the mean-field approximation is applied on the monomer level. The MT, on the hand, uses the probability distribution function of the polymer conformations, this means that the mean-field approximation is applied on the polymer level, i.e., chain correlations are considered. Moreover, representing a polymer chain by a continuous Gaussian chain implies a large amount of coarse graining and loss of molecular details. The MT is different from SCFT since the chains are represented explicitly and are self-avoiding. In the SCF approach, employing Gaussian chains, the chain conformations are not self-avoiding and obey random walks (RW) statics. Therefore, internal exclude volume constrains of one chain conformation are not properly taken into account. In the limit of low grafting density, the polymer properties such as size and density of the grafted chains follow a Random-Walk scaling behavior, while in the MT approach the size of polymer follow the proper scaling behavior associated with self-avoiding chains. For high molecular weight or denser polymer systems, where the screening length and critical overlap concentration are small compared to the polymer dimension and concentration, the polymers effectively follow RW statics and SCFT approaches become applicable [47].

#### Scheutjens–Fleer Lattice Theory

It is instructive to view the Scheutjens–Fleer Self-consistent (SF-SCF) theory as a lattice version of SCFT [48]. With the SF-SCF approach, the chain is presented as a first-order Markov chain on a lattice. Similar to SCFT, one can derive a now discrete version of the Edwards Equation, which governs the segment distribution function $G_{g}\left(\vec{r},s\right)$, from which one can compute the polymer density. Here, the variable “s” takes only discrete values, as it labels the segments of the chain. Thus, unlike in continuous Gaussian chain representation, the lattice chain of SF-SCF unambiguously identifies the segments. A graphical example of the lattice chain is shown in Figure 5. A comprehensive explanation of the SF-SCF theory can be found in [48,49].

Noteworthy, in the context of the present review, is the seminal work(s) by Israëls et al. and coworkers [50,51] the SF-SCFT approach was extended to weak PE brushes and they demonstrated the position-dependent nature of the degree of charge within a PE-brush. Similarly, they showed that the degree of charge within a PE-brush can be very different from that of an acid monomer in solution.

The SF-SCF has been very versatile in modeling polymer and polyelectrolyte brushes, and polymer adsorption. Attesting to this versatility is the fact that there are two papers in this special issue devoted to the SF-SCF approach. One describes the effect of the electric field on weak PE brushes [52] and the other investigates the pulling and pushing of a test chain in or out of an adsorbed polymer chain [53]. These two papers and references therein and two recent reviews [54,55] give further background on the SF-SCF method in describing polymers and polyelectrolytes near interphases and in solution.

### 3.3. Analytical Approaches

A common disadvantage of the above theories is that only numerical results can be obtained, which makes it sometimes difficult to recognize general trends. Following the work of de Gennes, various scaling theories have been formulated for polymer brushes [56]. By assuming very long polymers and, for example, also very densely grafted brushed, Edwards’ Equation can be simplified and approximated solutions can be obtained. This resulted in the identification of various scaling regimes of polymer and polyelectrolytes brushes [13,57]. Noteworthy are the Alexander–de Gennes model [58,59], analytical self-consistent theories [60,61,62], and the strong-stretching theories [37,40,63,64]. Here, one assumes that the polymer conformations are extended, which is valid only for densely grafted (and long) polymer brushes. The analytical theories for weak polyelectrolyte were based on original work for neutral and strong polyelectrolytes developed independently by Milner, Witten, and Cates [37] and Borisov and coworkers [65,66,67].

### 3.4. Simulations

Simulations provide exquisite molecular insights into both structure as well as kinetics. The acid–base chemical equilibrium mean can be introduced in simulations using constant pH or reaction ensemble simulations or lambda integration [68,69]. However, consideration of chemical reaction equilibrium combined with the large system size of a PE-brush makes these simulations computationally expensive. The numerical mean-field theories are computational inexpensive, allowing to study many different pH and salt conditions. The time-consuming nature of simulations hinders the survey of many different conditions. So far, simulations of weak polyelectrolytes have focused on polyelectrolytes in solution [68], where the number of polymers remains sufficiently low to perform simulations within a reasonable time. Simulations of weak PE-brushes are rare [70]. Simulations involving minimal approximations can be used to test the assumptions used by the mean-field theories [29]. With increasing computational power, simulations of the larger systems including tethered weak polyelectrolyte should become feasible.

## 4. Applications

Following the overview of theoretical methods given in the previous section, we would like now to present the reader to a number of relevant applications of them. Our intention is not to give an exhaustive or comprehensive enumeration of all the work done in the field, but rather to highlight salient examples of the different methodologies in the study and characterization of end-tethered weak polyelectrolytes. Although the selected examples focus mainly on grafted weak polyelectrolytes, we also briefly mention weak polyelectrolytes in solutions and systems that are not strictly end-tethered polymers, like gels and dendrimers. Our narrative briefly starts with weak polyelectrolytes grafted to planar surfaces, but rapidly moves onto the theoretical modeling of tethered weak polyelectrolytes to surfaces of different curvature and morphology, such as nanoparticles, stars and molecular brushes, and nanopores. We also present a number of systems with biological or biomedical relevance, which at first glance would not immediately be categorized as an tethered weak polyelectrolyte, but still share many common features with them. We also discuss some examples of chemical equilibria beyond acid–base reactions.


*Weak PE-Brushes: Charge Regulation*


The salient feature of weak PE-brushes is that the degree of ionization of the monomers within the layer is sensitive to the pH of the bulk solution. As expressed in Equation (Equation 3), the fraction of charged acid groups in ideal bulk solution is solely determined by the solution’s pH and by the equilibrium constant $K_{a}$. However, polyacid molecules are far from ideal conditions. Due to the chain connectivity, the charges of the monomers are correlated and the distribution and amount of deprotonation of the acid groups is influenced by both its position along the polymer chain and the local environment it experiences. This make their pH-dependent ionization much more complex than the ionization behavior of the titrable monomer in ideal bulk solution, given the coupling between the ionization states and the conformations of the polymer molecule. Indeed, titration curves of weak PE are smoothly varying functions of pH, because of the number of different ionizations states along the chain [14,16]. Moreover, in weak PE brushes, the anchoring of the molecule to a surface further influences the degree of charge because of the interactions the monomers experience within the polymer layer. In this way, different monomers experience different local environments depending on their location with respect to the surface. Thus, the degree of deprotonation becomes explicitly position dependent, as reflected in Equation (Equation 5).

Weak PE-brushes are able to optimize their charge state by (de)protonation depending not only on the solution’s properties (pH and salt concentration) but also on the local environment within the polymer layer. Let us illustrate this behavior for end-grafted polyacid chains (the extension to polybases is straightforward). At low pH values ($pK_{a}-pH\geq$ 2), the amount of deprotonated acidic monomers is very low and the polyelectrolyte layer is in a nearly uncharged, neutral state. Now, with increasing pH, the acid groups start to deprotonate and become charged. As the concentration of monomers is high inside the polyelectrolyte layer, deprotonation results in large electrostatic repulsions. To mitigate these increased electrostatic repulsions, the system has three main strategies: (1) to increase the electrostatic screening within the polymer layer by locally increasing the ions concentration; (2) to move the charged acidic monomers of the grafted polyelectrolyte further apart by stretching the polymer molecules further away from the surface and from each other; and (3) to remove the charge altogether, by shifting the acid–base equilibrium towards the uncharged species. The first possibility results in a loss of translation entropy of counterions as well as water molecules. As the system is incompressible, concomitantly with increasing the ion concentration, solvent molecules need to be exchanged, thus increasing the osmotic pressure within the polymer layer, which is unfavorable from a free energy perspective. The second option allows to move charges apart by swelling the polymer layer, but chain elongation results in loss of conformational entropy. Moreover, polymers are only finite-extendable, so this strategy can only help so far. Finally, the system can remove charges from the layer altogether, by protonating the acidic monomers. Through this mechanism of *charge regulation*, the total amount of charge can be reduced, but this comes with a chemical free energy penalty, dictated by the intrinsic equilibrium acid–base equilibrium constant. In weak PE-brushes, the possibility of regulating the degree of charge of the chains in response to changes in the local environment provides a feedback mechanism that allows tuning the balance between electrostatic, van der Waals, steric interactions, and molecular organization within the weak PE layer. The resulting amount of charge and degree of swelling arise from a non-trivial balance between chemical and physical interactions and molecular organization.

There is a large body of work on weak PE grafted onto planar surfaces. As our intention is to discuss the salient features of weak PE-brushes on curved and more complex geometries, we shall provide a brief overview of the methodologies that have dealt with planar weak PE-brushes. Derivation of analytical SCF approaches for a pH-responsive PE-brushes have been used to study the brush height, polymer profile, local degree of dissociation as a function of pH, ionic strength, and solvent quality [60,71,72,73]. Extending the SF-SCF theory to treat ionization equilibria in weak PE-brushes, Israëls et al. studied the effect of salt and pH of the bulk solution on the polymer layer thickness. They found that the thickness of a weak polyacid brush is a non-monotonic function of the salt concentration, whereas the thickness of a strong PE-brush is a continuously decreasing function of salt concentration [51]. The solvent quality was found to dominate the properties of the brush for pH values for which the weak PE are effectively neutral [74]. The interplay between polymer hydrophobicity and conformational properties was also examined with a numerical SF-SCF theory [75]. Witte et al. employed the numerical SCF formalism to study of the effect of grafting density on the height of a weak PE-brush [43].

Das’ group expanded on the strong stretching theory for pH-responsive PE-brushes [73] to account for the effect of excluded volume interactions in weak poly-acidic planar brushes immersed in an electrolyte solution [64]. The molecular theory (MT) developed by Szleifer and coworkers was found to quantitatively predict the structure of end-grafted weak PE as a function of bulk solution’s pH, the ionic strength, and the density of polymer, as compared with experimental observations [24]. The methodology was extended to study the balance between electrostatic, van der Waals hydrophobic interactions, steric interactions, and molecular organization in grafted weak PE [76].

The single chain in mean-field model (SCMF) developed by Léonforte and Müller for planar weak PE-brushes is an interesting theoretical approach, where one explicitly simulates the chain interacting with the external field that results from all other chains and small ions in the system [45]. They used a discretized Edwards Hamiltonian in combination with the Gaussian chain model. It is a hybrid between explicit particle coarse-grained simulations and standard mean-field models. A salient feature of the model is that it allows to predict kinetic effects.

Commonly, the dielectric medium of a PE-brush is assumed to be that of water and effects due to variation in relative dielectic constant are rarely considered, i.e., $\epsilon_{r}\left(r\right)\approx \epsilon_{w}$. Worth mentioning are the SCF calculations performed by Kumar and coworkers to analyze the effect of a varying dielectric function on the charge of a planar polyacid brush as a function of added salt and pH of the bulk solution [44]. Likewise, MT and SCMF studies have also considered the effect of both varying dielectric constant and Born solvation energy on the charge and structure of polyacid brushes [45,77]. However, they show that under good solvent conditions and intermediate grafting densities the assumption of $\epsilon_{r}\left(r\right)\approx \epsilon_{w}$ suffices.

The references provided in the above paragraphs show that the optimization of charge state in planar weak PE-brushes by (de)protonation depends not only on the solution pH but also on the local environment. Next, we will see that surface curvature, polymer topology, the presence of multivalent counterions, or poor solvent conditions can add further complexity to the ionization of weak PE.

### 4.1. Nanoparticles

Interest on nanoparticles (NPs) end-tethered with weak PE molecules stems from the possibility of taking advantage of how the acid–base properties of the coating ligands change when they are subjected to different environmental conditions. Mechanisms of pH-responsive NPs encompass pH-dependent swelling and dissolution, aggregation, drug dissociation and release, pH-labile linkers, pH-sensitive conjugates, and pH-triggered phase transitions. When designing smart materials of this kind, the curvature and morphology of the NP play a central role in its surface modification and the functionality of the final nanocomposite, as they determine the available volume to the molecules grafted to its surface and how it changes moving away from the interface. Indeed, this volume can increase, as it does in curved convex NPs like spheres and cylinders; it can decrease, as in curved concave objects like nanopores and nanochannels; or it can remain constant, like with planar surfaces. Immobilizing a molecule onto a surface affects its translational and conformational entropy with respect to the free molecule in solution in a way that greatly depends on the geometric constraint imposed by the surface. The penalty this imposes ultimately modulates the molecular organization of the surface species as well as the physical interactions and the chemical state of all the species in the system.

#### 4.1.1. Charge Regulation & Curvature

The interplay between charge regulation in weak PE brushes at surfaces of different morphology and curvature was systematic evaluated employing MT calculations [18]. The authors studied the effect of surface geometry and solution pH and ionic strength on the acid–base equilibrium of weak polyelectrolytes tethered to planar, cylindrical, and spherical surfaces. Surface geometry was found to play a dramatic role in acid–base equilibrium as well as in the molecular organization of the coating layer. Interestingly, the position-dependent degree of dissociation and pH are significantly affected by the curvature of the surface, exhibiting large variations even within a few nanometers.

In a later publication, combined titration experiments with MT calculations investigated the acid–base equilibrium of the terminal carboxylic group in gold NP (Au-NPs) of different sizes coated with self-assembled monolayers of mercaptoundecanoic acid (MUA). [26] The schematic representation of the system is portrayed in the the upper right panel of Figure 1. In that same figure, panel A shows the fraction of charged/dissociated MUA on the NPs as obtained from experiments, and from the MT calculations as well as using the simple Henderson–Hasselbalch equation for free MUA molecules in ideal solution (i.e., Equation (Equation 3)). Clearly, that assumption of ideal solution behavior for the MUA tethered to the NPs is not applicable. To further characterize the dissociation of the acid groups, the authors compared experimental values with MT predictions of the apparent $pK_{a}$, defined as the pH value for which the degree of dissociation is 0.5. They found that its value is significantly higher than the $pK_{a}$ of MUA in solution (~4.8) and that it strongly depends on the NP’s diameter and the ionic strength of the solution (Figure 1, panel B). This can be rationalized based on the available volume for the grafted MUA: given that the distance between acid groups depends on the NP size, for the same surface coverage of ligands, the average distance between head-groups of MUAs decreases with NPs’ increasing diameter. Therefore, as the size of the NP increases, it is harder to ionize the carboxylic head-groups. The authors also found an excellent agreement between experiments and MT calculations when varying the cation size, by evaluating different organic salts in solution (Figure 1, panel C). MT calculations also enabled to rationalize experimental observations on the charge state of MUA chain on NPs of mixed geometries, such as gold nanorods and nano-dumbbells, for which it was observed that regions of the particle surface with different curvature become charged at different pH values of the surrounding solution [78].

#### 4.1.2. Conformational Properties of Grafted Weak PE

The charge distribution on weak polyelectrolytes is a key factor, dictated by the local environment in terms of physical interactions and molecular organization. Particularly, charge regulation provides a way to modulate interactions between neighboring chains and also, the conformations they adopt. There is quite an abundant body of theoretical work on weak PE in aqueous solutions, focusing on the interplay between charge regulation and fluctuations in the conformational properties. For this topic, molecular simulations are very well suited, as they can provide direct insights and molecular detail on the behavior of the chains in different conditions, and several studies employing hybrid or coupled MD and MC schemes have been performed in the last couple of years in this direction [68,79,80,81,82,83,84].

For weak PE brushes on NPs, the presence of the surface also critically affects the molecular organization of the chains. Theoretical methods, including scaling theory [85], mean-field approximation approaches [86], numerical SCF methods [87], and SF-SCF calculations [88,89,90], have contributed greatly to what we know about the effect of bulk pH, salt concentration, and surface density of chains on the conformational properties and the distribution of charge of weak PE grafted to curved surfaces. In a work combining grand canonical titration Monte Carlo simulations of coarse-grained weak PE chains end-tethered to a spherical NP in a salt solution, Barr and Panagiotopoulos investigated the effect of solvent quality and pH of the solution on the ionization state of the monomers and the phase behavior of the grafted layer [70]. The authors found that increasing the attraction strength between the monomers (ϵ∗) decreased the charge on the polymers (Figure 2, panel A). Doing so, polymer beads local density increases, which downregulates the monomers’ ionization in order to reduce electrostatic repulsions. For pH − pK${}_{0}$≥ 1, a relatively abrupt decrease in charge state is observed, which is driven by homogeneous layer instability for high enough values of ϵ∗. Four distinct polymer morphologies were observed: a homogeneous brush (HB), stripes (S), micelles (M), and non-aggregated chains coexisting with micelles (M+NA) (Figure 2, panel B). Similar morphologies have also been predicted by MT calculations for planar weak PE brushes in different solvent and pH conditions [76]. The phase diagrams for different grafting densities were also computed (Figure 2, panel C).

#### 4.1.3. Interactions between Weak PE Coated NPs

The possibility of modulating the strength and nature of the interactions between NP functionalized with ionizable PE is a key factor in building more complex structures with specific envisioned functionalities, such as sensors, switches, and motors. MT calculations were used in several publications addressing this topic. Popov and coworkers used the MT to characterize the interactions between two spherical NPs coated with short polymer chains containing an ionizable end-group immersed in aqueous salt solution [91]. They found the interacting NPs to mutually influence the charge regulation of the end-groups, inducing an asymmetric distribution of charge, which conferred a preferred directionality in the NP–NP interactions.

In another work, MT was used to predict and rationalize experimental results on the stability of citric acid-coated/PEGylated iron oxide NPs under different pH and ionic strength conditions [92]. More recently, MT calculations provided the design criteria to achieve dispersion stability of NPs coated with polyacrylic acid (pAA), poly acrylamido-2-methylpropane sulfonate (pAMPS), or an alternating copolymer of the two (pAA-a-AMPS) under brine-like conditions (high pH and salinity of the solution) [93]. The authors explored the effects of surface density (Figure 3, panel A), NP core size (Figure 3, panel B), solution’s salt concentration, and pH (Figure 3, panels C and D, respectively). Very recently, these MT predictions were used as guidelines to synthesize and successfully achieve dispersion stability of iron-oxide NP coated with pAMPS in brine solutions, demonstrating the possibility to use theoretical calculations to rationally design stable NP dispersions [94].

#### 4.1.4. Interactions between Weak PE-Coated NPs and Surfaces

Saito et al. modeled the adsorption of heterogeneously charged NPs onto oppositely charged surfaces, considering a simple distribution of $pK_{a}$ values for the anionic functional groups. The authors extended the surface complexation approach to combine a numerical solution of the mean-field Poisson-Boltzmann equation with surface complexation reactions to calculate the electrostatic free energy of the adsorbed particle [95]. The work focused on the mutual charge regulation between the NP and the surface, exploring the effects of pH, salt concentration, and chemical heterogeneity of the NP. In a more recent publication, Nap and coworkers employed MT calculations to study the adsorption of NPs coated with neutral polymers end functionalized with surface acid groups onto both negatively and positively charged surfaces, as a function of pH and salt concentration, polymer coating, and NP size [96]. Proximity to the charged surface was found to induce an asymmetric charge distribution on the NP: the acid-coated NPs downregulate their amount of charge, whereas upregulation is observed for NPs close to a positively charged surface. In both cases, charge regulation leads to an increase in the adsorbed amount of NPs. The authors found a non-monotonic adsorption as a function of pH.

#### 4.1.5. Interactions between Weak PE + NPs and Surfaces

We finish this section by briefly departing from grafted weak PE to consider the interactions between PE chains in solution and oppositely charged macroions or NPs.

The topic of PE and oppositely charged macroions (or NPs) complex formation has been studied by means of MC simulations by several groups [97,98]. Ulrich and coworkers published a somewhat dated but still accurate review, analyzing the formation and structure of complexes based on the principle of electrostatic complexation, with a focus on results from computer simulations [99]. Applying MC schemes, Stoll and coworkers found that the adsorption/desorption limit and the number of monomer in contact with the macroion surface depended on the salt concentration and pH of the solution and on the macroion surface charge density, as summarized in Figure 4. It was observed that the interactions between the weak PE monomers and the charged macroion profoundly affected the charging state of the PE chain, promoting chain ionization. Increasing salt concentration increases the electrostatic screening, therefore the attractive interactions between the PE and macroion, but at the same time it promotes PE ionization degree. The best conditions for the formation of strong complexes were found to be at intermediate salt concentrations. Later publications by the group continued exploring this topic expanding to polyampholyte chains [100] and solutions containing multivalent ions [101].

The adsorption of PEs on oppositely charged surfaces has also attracted considerable research interest in the last decades. This process is mainly governed by the interplay between the electrostatic attractions, the loss of polymer conformational entropy upon the adsorption, and the steric repulsion between the polyelectrolyte segments, all of which are modulated by the surface curvature. Strong PE adsorption has been studied by SF-SCF onto planar surfaces [102], and more recently, by MT calculations onto spherical and cylindrical NPs and nanopores [103]. The underlying mechanisms were nicely summarized by Winkler and Cherstvy in a recent review [104].

For adsorbing weak PE, studies have been more scarce. In those systems, the adsorbed amount of is also controlled by the pH and the ionic strength of the solution. For planar surfaces with variable charges, early SCF calculations showed that both the surface and the weak PE chains are able to “titrate” each other in a non trivial way [105]. More recently, Tong studied the adsorption of weak PE onto charged NPs applying numerical SCF theory. The author analyzed the dependence of the thickness of the adsorbed layer on the charge density and the radius of the NP, and the bulk monomer charge fraction. The curvature effect of the charged sphere on the degree of charge compensation was also examined [106].

### 4.2. Star and Bottle-Brush Weak PE

Star and bottle-brush polymers can be regraded as macromolecular assemblies consisting of chains end-tethered to spherical and cylindrical cores in the limit of very small core radius.

Wolternik and coworkers employed the SF-SCF theory and analytical approaches to explore the equilibrium conformations and charge state of annealed star-branched PE, as a function of the number of arms, pH, and the ionic strength [107]. Later on, theoretical studies were extended to analyze the conformational properties of flexible-chain ionic dendrimes (also referred to as starburst polyelectrolytes) in dilute solutions. Numerical SCF calculations were also performed on weak PE dendrimers in solution, reporting charge regulation in response to the density of polymer monomers, counterions, pH, and salt ions [108].

Going into further detail on the conformational properties of pH-sensitive starlike polyelectrolytes, Uhlik et al. compared results from hybrid MC simulations (reaction ensemble, implicit solvent) with numerical SF-SCF calculations and analytical approaches in salt-free solutions [109]. Figure 5 shows the titration curves (that is, the average degree of dissociation of the star weak PE as a function of pH) obtained with both methodologies. Results are compared with experimental data and with the curve for a single monomer in ideal bulk solution. It can be seen that all curves are shifted to higher pH values with respect to the ideal curve, given the non-ideality of the dissociation of ionizable monomers due to chain connectivity. This effect becomes more significant with increasing number of arms. Both methods show a decrease in the degree of ionization due to the crowding in the central part of the star. However, the authors observed a rise of ionization at the periphery of the star in the MC simulations that was missing in the SCF results. This discrepancy was attributed to underestimation of intra-arms interactions within the mean-field approximation. This work is very thorough in critically analyzing the features of the different methodologies. While MC simulations directly sample configurations of the stars, in the SCF calculations individual polymer conformations are never directly available. On the other hand, the SCF method is computationally much less intensive, but it is based on the mean-field approximation which breaks down when correlations become important. MC simulations can in principle provide exact results for a given model but are computationally expensive and limited by the statistical accuracy attainable at reasonable computational cost.

Interacting stars or stars interacting with surfaces were more recently studied by Rud, Birshtein, and coworkers using SF-SCF approaches [110,111]. The conformations of two interactive weak PE stars with amphiphilic segments change significantly as a function of the distance between them, from a two-phase quasi-micellar conformation for the individual stars while being far apart to a quasi-micelle when the stars touch each other [110]. The authors also investigated the effect of salt concentration and pH. Later on, the study was extended to a hydrophobic weak PE star interacting with a hydrophobic surface [111]. It is shown that the hydrophobic core of the star provides a driving force for the quasi-micellar star to adsorb onto the surface.

Molecular brushes or bottle-brushes comprised of multiple ionizable side chains tethered to the main chain (backbone) also recently became the focus of several SFC works. Borisov and Zhulina developed a mean-field theory to study the solution properties of strong PE molecular brushes [112]. They observed a strong localization of counterions within the polymer layer, accompanied by full extension of the backbone. This stretching was found to decrease with increasing salt concentration. In a later publication, analytical mean-field theory and numerical SF-SCF modeling were used to study the competition between long-range Coulomb repulsion with short-range solvophobic interactions in pH- and thermo-responsive molecular brushes [113]. The authors found that this interplay between the physical and chemical interactions in the system lead to complex patterns in the intramolecular self-organization of the molecular brushes. Particularly, they observed the formation of *intramolecular micelles* appearing as pearl necklace-like structures, in which multiple dense nanosized domains formed by weakly ionized collapsed side chains while being stabilized by a fraction of strongly ionized ones extending towards the solution (Figure 6). Salt concentration was found to have a profound impact on the intramolecular nanopatterning.

### 4.3. Biomacromolecules

In this section, we would like to provide the reader with salient examples of the different theoretical methodologies described in Section 3 applied to a number of biological systems and biological macromolecules that even though they might not be immediately categorized as tethered weak PE, still share many features and behaviors with them.

#### 4.3.1. Neurofilaments

Neurofilaments (NFs) are the main cytoskeletal constituents of myelinated axons [114]. They can be seen as weak PE cylindrical brushes with a radius of the backbone much smaller than the brush thickness. NFs are composed of three molecular weight subunits: NF-L (low), NF-M (medium), and NF-H (high). These subunits self-assemble to form a 10 nm thick filament backbone, with protruding side-arms formed by their C-terminus tails, which are intrinsically disordered and highly charged proteins. The interactions between neighboring NFs dictate the interfilament distance as well as the structural cohesion between filaments. The networks of side-arm-mediated NF assemblies play a key role in the mechanical stability of neuronal processes [115]. Given the presence of titrable amino acids, NFs have also inspired the design of stimuli-responsive nanoconstructs formed by NPs grafted with intrinsically disordered proteins [116,117].

Experimental work from Beck and coworkers shed light on the importance of explicit amino acid sequence and acid–base properties when modeling the side-arm-mediated interactions between NFs [114]. In that line, Zhulina and Leermakers conducted the first theoretical study of NFs brushes with amino acid resolution [118]. Combining numerical SF-SCF approaches with coarse-graining of the NF tails, the authors explored the equilibrium structure and conformational properties of an individual NF under physiological pH (pH = 7) and relatively low ionic strength (0.01 M of a monovalent salt). Tails corresponding to the NF-H, NF-M, and NF-L subunits, each with its own size and sequence (N${}_{H}$ = 607, N${}_{M}$ = 504, and N${}_{L}$ = 142), were tethered to the backbone with the stoichiometric ratio H/M/L = 2:3:7. The authors characterized the structure of the NF brush, finding a very marked and inhomogeneous distribution of each tail in space. They also explored the effect of phosphorylation, by which each transfer of a phosphate from ATP changes the charge of the KSP repeat in subunits NF-M and NF-L from +1 to −1. The ionization of these repeats triggers a major relocation of the H-tails from the backbone of the NF to the periphery of the brush. This leads to an increase in the height of the NF brush and it ultimately translates into an increased interfilament distance. In later publications, the authors also explored the effects of ionic strength and pH [119], protein composition [120], subunit sequence [121], and NF–NF interactions [122]. Structural and conformational dynamics of NFs have also been explored using MD [123] and MC simulations [124,125,126,127].

#### 4.3.2. Aggrecans

Aggrecans are proteoglycans, proteins that are modified with large carbohydrates, and they are one of the most abundant components of cartilage [128]. Their main function of is to provide a hydrated gel structure that resists compressive forces, granting the cartilage with load-bearing and lubrication properties. An aggrecan molecule contains about 100 chondroitin sulfate-glycosaminoglycans (CS-GAGs) chains covalently bound to a 300 kDa linear core protein chain (backbone) that has a contour length of 400 nm. Each repeat unit of the GAG side chains has one sulfonic and one carboxylic group. Therefore, similarly to neurofilaments, aggrecans can also be described as weak PE cylindrical brushes, in which both the backbone and the side chains are ionizable. Under physiological pH conditions, the side chains are mostly negatively charged, with significant counterion confinement within the layer, that is responsible for the extremely high osmotic swelling pressure of cartilage. Theoretical investigations exploring the interactions of GAG chains and aggrecan at the molecular level are not abundant. Of particular interest are the MT calculations employed to study the behavior of an isolated aggrecan molecule as well as the interactions between two aggrecans [129]. The authors modeled them as nanometer-sized cylindrical surfaces tethered with weak polyelectrolytes under different pH and ionic strength conditions. Interactions between two aggrecan molecules were found to be strongly repulsive, increasing with acidity and decreasing with salt concentration. In turn, the interdigitation of two aggrecan molecules is also strongly affected by the salt concentration and the solution’s pH. At high pH values, the charging of the side chains increases, increasing electrostatic repulsions between polymers, thus leading to a lower interdigitation. The authors rationalized their theoretical findings in the context of the known lubrication properties of cartilage.

#### 4.3.3. Viral Capsids and Gene Vectors

Viruses are systems that have developed distinct mechanisms to ensure the proper encapsidation and stabilization of their highly negatively charged genomes. Electrostatic interactions play a major role in capsid stability and assembly, as well as genome packaging and ejection [130]. In this context, any mechanism that might affect the charge state of the system and Coulomb interactions (either by charge regulation, electrostatic screening, among others) is expected to have deep effects.

The coat protein of positive-stranded RNA viruses often contains a positively charged tail that extends toward the center of the capsid and interacts with the viral genome. The electrostatic interactions between the tail and the RNA have been postulated as a major driving force in virus assembly and stabilization. In this context, combining experimental and theoretical work, Peng and coworkers examine the correlation between electrostatic interactions and amount of RNA packed in the tripartite Brome Mosaic Virus [131]. The authors used a SF-SCF scheme, treating electrostatics with the nonlinear Poisson–Boltzmann equation. The assembled capsid was treated as a spherical cavity grafted with N-terminal tails, which were modeled as flexible strong PE. The capsid was considered impermeable to the N-terminal tails and RNA but permeable to water and small ions. The authors computed the free energy of a theoretical virion due to confinement of the RNA and interactions among the RNA, the N-terminal tail, and ions, as a function of the number of charges in the peptide tails. They found that the free energy varies non-monotonically with RNA length, with a minimum that corresponds to the length of RNA around which the assembled capsid is most thermodynamically stable. This is a good example of application of theoretical modeling, however it did not include charge regulation or weak PE.

Viruses and virus-like particles often carry large surface charges, making surface charge density an important system parameter. Examples of theoretical studies incorporating the acid–base equilibrium of the titrable amino acids are scarce [132,133]. Worth mentioning is the work conducted with MT calculations to quantify the effects of pH and salt concentration on the charge regulation of the bacteriophage PP7 capsid [134]. The overall charge of the virus capsid was found to emerge from a complex balance between the ionization of the titrable amino acids, the electrostatic interactions between them, and counterion release. The amino acids in the capsid regulate their charge state as to avoid large electrostatic repulsions, governed by the pH and salt concentration of the solution. The authors observed a transition from net-positive to net-negative charge depending on the solution’s pH. Moreover, they also found a buffering behavior under physiological conditions of pH and ionic strength.

#### 4.3.4. Proteins

Proteins can be regarded as weak PE with many titratable groups and various $pK_{a}$ values. Given the possibility of regulating charge in titrable amino acids, solution conditions as well and local environment have a major influence on electrostatic interactions and with that on their structures, stability, assembly, solubility, and function. Simulations of charge regulation in proteins were reviewed by several authors [68,135].

Protein adsorption is a technologically relevant phenomenon, with applications in protein purification and separation. It is particularly important when engineering materials for biomedical applications, as serum proteins and cells tend to adsorb onto their surface, forming a “protein corona”, that imparts a biological identity to the material, and it determines its in vivo distribution, systemic clearance, downstream biological effects, and reactivity in physiological media [136]. During adsorption, proteins interact with the substrate through a combination of electrostatic, van der Waals, hydrogen-bond, and hydrophobic interactions. On charged surfaces, the electrostatic interactions are the dominant force and the amount of adsorbed protein will depend on the net charge of the protein and the substrate. Several experiments have shown that protein adsorption can occur on bare or polyelectrolyte-modified surfaces even when the charges of the protein and the surface are of the same sign. It has also been reported that the adsorbed amount of proteins displays a maximum at pH values near the isoelectric point of the protein. There is an ongoing question of whether charge regulation or patchy charge distribution is the dominant mechanism when the overall charges on both the protein and the substrate have the same sign. In the context of charge regulation, the presence of the charged surface groups induces a charge of the opposite sign in the protein by shifting the acid–base equilibria of its amino acid. At the same time, the charge-patch mechanism is based on “charge patches” on the protein that allow optimizing protein–substrate interactions by changing protein orientations. De Vos and coworkers developed a model based on SF-SCF theory for proteins adsorbing onto planar surfaces modified with like-charged PE brushes [137]. They found that both mechanisms operated simultaneously, although charge patchiness did so to a lesser extent. Still, both charge anisotropy and charge regulation effects promoted protein adsorption at the “wrong” side of the isoelectric point, and their effects were found to be additive. The question of charge patches versus charge regulation was also addressed by Boubeta and coworkers for electrostatically driven protein uptake [138]. The authors employed MT calculations to study the adsorption of different proteins onto charged surfaces, exploring the effects of pH, salt concentration, protein orientations an the protein–surface interactions. They also found both mechanisms to play their part at low ionic strengths, whereas the charge-patch mechanism dominates at high ionic strength. The contribution of charge regulation showed to be insensitive to protein orientation under all conditions. MT calculations were also employed to study the effects of charge regulation on protein adsorption in pH-responsive hydrogels [139,140].

### 4.4. Beyond Acid–Base Equilibrium

So far we have focused on weak PE-brushes tethered to curved nanosystems. We would like to include in this review a broader perspective of chemical reactivity in terms of the different types of chemical reactions and physical interactions that can also be studied employing theoretical methodologies as the ones outlined above. As long as it is possible to describe the process in terms of a chemical equilibrium, the characterization given in Section 3 still applies. Examples of such processes include PE brushes where the segments can be oxidized or reduced (redox reactions), brushes with ion binding or ion complexation capabilities, brushes with ligands that recognize specific molecules in solution, and polymers that can reversibly bound/unbound to the substrate. Between them, we would like to highlight ion binding and ligand receptor binding, as they are very relevant in biological scenarios.

Using the above described methodologies to describe chemical reactions can also be applied to physical processes such as hydrogen bonding and ion-pairing. For example due to strong short ranged electrostatic attractions, oppositely charged species will localize in close proximity and form ion-pairs. The formation ion-pairs or ion-binding although of physical origin can be described effectively in terms of a chemical reaction chemical equilibrium reaction between the unbound ions and the ion-pair.

#### 4.4.1. Ion Binding

In monovalent salt solutions, the height of strong PE brushes decreases monotonically with the salt concentration, whereas for multivalent salt solutions, a drastic collapse is observed, resembling the swelling behavior for a neutral brush in a poor solvent. Examples of publications of how di- and trivalent ions induce such a sharp decreases in brush thickness as a function of their valence and concentration include Non-local DFT [141], Langevin dynamics simulations [142], and combined experimental and coarse-grained MD simulations [143]. For planar polyelectrolyte brushes of poly(styrenesulfonate) in contact with solutions containing trivalent ions, the observed structures (both from the MD simulations and AFM) strongly resemble those predicted for polymer and polyelectrolyte brushes in poor solvents, and were rationalized by the ability of multivalent ions to form intrachain and interchain bridges as well as nearest neighbor condensation [143]. Multivalent ions were also found to have a strong impact on the lubricating properties of opposing polyelectrolyte brushes sliding against each other [144].

For weak PE in solution, Rathee and coworkers found that divalent ions induced a coil-to-globule transition. Using hybrid MC-MD simulations, the authors observed the weak PE structure to collapse, similar to that observed in strong polyelectrolytes, even though the ionization degree was higher in comparison to the monovalent salt, where a swelling of the polyelectrolyte was observed instead [80]. Employing a simple box model (Alexander–de Gennes model), Birshtein and Zhulina investigated the effect of ions valency on the equilibrium behavior of weak polyacid chains grafted to a planar surface immersed in an infinite reservoir of water, H${}^{+}$, OH${}^{-}$, and ions of different valency [145]. It is worth noting that the formation of complexes between negatively charged monomers and positively charged polycations was not considered, the interactions between them was assumed to be only through electrostatic interactions. The authors found a non-monotonic dependence of the brush thickness with the ionic strength and counter ion valency. In a later publication, using a similar scaling framework, Zhulina and Borisov expanded the study to encompass two types of multivalent salt ions, namely, co- and counterions [146]. Here, again salt ions were not consider to explicitly bind to the charged monomers and the mobile ions and the polyions interact through a self-consistent electrostatic field associated with the grafted layer. For both types of multivalent ions, the authors observed that the layer thickness passes through a maximum at moderate salt concentrations, and that the degree of ionization of the monomers increases monotonically with the ionic strentgh of the solution. However, in the case of multivalent counterions, the scaling exponents became sensitive to the counterion valency. The authors attributed their results to multivalent counterions being more effective in screening the electrostatic interactions in grafted weak PE at low salt concentrations.

The preceding scaling theory examples illustrate the mechanism of charge regulation in weak PE driven solely by the electrostatic interactions between the ionizable monomers and the multivalent counterions. In the same line, we can mention a more recent work using MT calculations in which single-stranded DNA monolayers were modeled by planar polyacid brushes immersed in solutions containing different pH, salt concentration, and cation identity (Na${}^{+}$, Mg${}^{2+}$) [147]. Modeling included the acid–base equilibrium of the monomers, but no explicit ion-binding. The authors found that the degree of protonation can change dramatically even when the bulk pH is kept fixed simply by changing the surface coverage of the polyacid and the bulk salt concentration. This is a direct manifestation of the monomers shifting the acid–base equilibrium in order to minimize the system’s free energy. Decreasing salt concentration while keeping the bulk pH constant, the polymer chains stretch away from the surface to decrease electrostatic repulsions until they reach their full length. Reducing salt concentration past this point results in prohibitively high electrostatic repulsions, so the system minimizes them by shifting the acid–base equilibrium of the monomers towards the neutral and protonated state. Moreover, for conditions in which monomers are almost completely protonated, an expulsion of Na${}^{+}$ ions was observed. Now, when considering Mg${}^{+2}$ solutions, these effects were reduced. This was attributed to the multivalent cations being more effective in compensating the polyacid charge with a lower steric and entropic cost, reducing the benefit in the free energy by shifting the acid–base equilibrium.

Another possibility to regulate the charge state of the monomers is through counterion condensation or ion binding. This ion-binding reaction can be treated similarly to an acid–base equilibrium, as described in Section 3. Starting again with the example of carboxylic acid binding to Na${}^{+}$ ions in solution, the chemical reaction can be described as $A^{-}+Na^{+}⇌ANa$ and characterized by a chemical equilibrium constant and an extent of the reaction (or fraction of bound monomer) in analogy to $K_{a}$ and $f_{A^{-}}$. MT calculations were used to study sodium ion binding to NP coated with polyelectrolytes in high salinity solutions [93]. As mentioned above, the authors focused on exploring the conditions that would render stable solution of NP modified with either polyacrylic acid (pAA), poly acrylamido-2-methylpropane sulfonate (pAMPS), or an alternating copolymer of the two [93].

Counterion condensation is often more common for multivalent ions than for monovalent ions. This condensation leaves the polyelectrolyte with a low effective charge. At the same time, multivalent ions are also capable of condensing onto multiple polyelectrolyte monomers, which could induce bridges between chains and an alternate type of attraction that may drive polyelectrolyte brush collapse. Let us consider the example of divalent cations, such as Ca${}^{2+}$ or Mg${}^{2+}$. The possible reactions would then be
(9)$$\begin{array}{ccc} & \begin{array}{c} A^{-}+Ca^{2+}⇌ACa^{+} \end{array}, & \end{array}$$
(10)$$\begin{array}{ccc} & \begin{array}{c} 2A^{-}+Ca^{2+}⇌A_{2}Ca \end{array}. & \end{array}$$
each of them characterized by an equilibrium constant and a fraction of bound, either 1:1 or 1:2.

The 1:1 binding of Mg${}^{2+}$ was explored within the MT framework for ssDNA oligomers grafted onto planar surfaces [148]. The results exhibited the expected collapse of the layer with increased ionic strength at low grafting density and the re-entrant phenomena of stretched chains with increased ionic strength at high grafting density. In turn, 1:1 binding of Ca${}^{2+}$ to micelles composed by carboxylic acid end-standing Pluronic P85 block copolymers was studied both experimentally by differential potentiometric titration measurements and by SF-SCF theory [149]. The authors included both acid–base equilibrium and ion condensation reactions into the model, and found that Ca${}^{2+}$ ions bind both electrostatically and specifically. The specific binding between Ca${}^{2+}$ and carboxylic groups in the corona of the micelles was calculated to be pKCa 1.7 for the condensation reaction (Equation (Equation 9)).

More recently, MT calculations were used to investigate the structural changes of polymer-coated NPs with copolymers of poly(acrylic acid) (pAA) and poly acrylamido-2-methylpropane sulfonate(pAMPS) in the presence of Na${}^{+}$ and Ca${}^{2+}$ ions [30]. The mechanism of Ca-binding to the end-tethered pAA layers was explored, considering both 1:1 and 1:2 binding (Figure 7). The authors found that end-grafted PAA layers collapse in solutions containing sufficient amounts of Ca${}^{2+}$ ions (Figure 7, panel B), due to the formation of calcium bridges between two acrylic acid monomers and one calcium ion. This is strongly dependent on the pH as well as divalent and monovalent salt concentrations, and was rationalized by analyzing the fraction of bound monomers to protons, Na${}^{+}$ ions, and Ca${}^{2+}$ ions (Figure 7, panel C). Later publications expanded on the effect of curvature in cylindrical NPs [150], and explored the feasibility of engineering gating devices based on pH-responsive nanochannels as a result of Ca${}^{+2}$ bridging [151].

#### 4.4.2. Ligand–Receptor Binding

The binding between a ligand and its receptor is an important process in many biological systems, including immune reactions, signaling, opening of ion channels, and gene activity. Ligand–receptor (L–R) binding is a chemical equilibrium process that can be as well characterized as was done for acid–base equilibrium in Section 3. Furthermore, similarly to charge regulation in the acid–base case, ligand–receptor binding can be shifted towards the dissociated species or the bound pair depending upon the conditions of the environment and the intrinsic binding constant, which is pair-specific. The high specificity and high affinity ligand–receptor binding makes this interaction very attractive for a variety of applications, particularly in the biomedical field. Some examples include the binding of magnetic particles to cells for imaging tracking; targeted drug delivery to cancer cells; or ligand-binding assays that use ligands to detect, target, or measure a specific receptor.

Attaching the ligand to a surface modified with polymer to engineer more complex nanoconstructs raises the question of how would this grafting process affect the binding equilibrium between the ligand and receptor. In order to provide insights to this question, MT calculations were used to investigate L–R interactions in polymer brushes on planar surfaces and spherical NPs [152,153,154]. A common observation was the existence of a non-monotonic relation between L–R binding and the surface coverage of ligands. The presence of the maximum is due to the ability of the polymer-bound proteins to optimize binding while dispersing the ligands in space and minimizing lateral repulsions [154]. This also allows tuning the orientation of the bound proteins by properly engineering the layer at the molecular level. The effect of confinement on L–R interactions and binding was also studied for proteins binding inside modified nanopores by means of MT calculations [155].

In biomedical research, a key challenge is to design and engineer a system with the ability to specifically target cells and tissues. MT calculations were used to analyze how lipid composition and charge regulation affect the interactions between coated nanomicelles with membrane receptors and the binding between them [156]. Three different model membranes were explored: (1) a neutral lipid membrane with overexpressed receptors, (2) a membrane with negatively charged lipids (no receptor), and (3) membranes with both overexpressed receptors and negatively charged lipids. The micelles were coated with a binary mixture of short neutral polymers and polybases. These polybases, which can become positively charged, were in turn modified with a functional end-group for specific binding to the membrane-receptors (Figure 8, panel A). The authors found that combining ligand–receptor binding and electrostatic interactions lead to a very pronounced segregation between the neutral and charged polymers on the micelle surface (Figure 8, panel B). At the same time, they observed that both the polybases on the micelle and the lipid molecules in the membrane mutually regulated their charge state (Figure 8, panel C). As can be seen in the left panel, given the crowding of polybases grafted to the micelle, their fraction of dissociated groups is less than the expected 0.5 (as pH = pK${}_{a}$), with the only exception of two points in the region closer to the lipid layer. In these regions, the fraction of charge is enhanced due to the electrostatic attractions with the negatively charged lipids. The corresponding charge enhancement is also observed for the lipid molecules in the membrane (Figure 8 C, left panel). The interplay between L–R, electrostatic, and steric interactions, modulated by the curvature of the micelle, results in a much larger binding (blue line, Figure 8, panel D) than the one corresponding to the sum of the independent contributions (red and purple lines, Figure 8, panel D). Even more interesting is the fact that there are regions of interaction between the micelle and the membrane where combining two effective repulsions leads to an overall attraction. This work provides a good example on how to improve targeting by using multiple physical and chemical interactions. This polivalency strategy is becoming a powerful tool to engineer multivalent NPs with enhanced selectivity and binding. In this line, it is worth mentioning the MC simulations by Curk and coworkers, where they showed that it is preferable to exploit multivalency and target a specific receptor profile rather than aiming for fewer but stronger binding interactions [157].

### 4.5. Artificial Nanopores

So far we have focused on the theoretical understanding of polymer tethered to flat and convex surfaces. One could imagine that it is more challenging to graft polymers to concave surfaces such as those of nanopores or nanochannels (the two terms are often used interchangeably). Nevertheless, the advance of nano-fabrication and polymerization techniques has allowed the application of polymers and macromolecules to functionalize nanopores for sensing and gating purposes. From synthetic polyelectrolytes to polypeptides to oligonucleotides, a wealth of polymeric materials have been designed and employed as functional coating layers for artificial nanopores. Many applications of nanopores and porous materials are based on the unique transport properties of mass through nanoconfined geometry. For example, hydrophobic nanochannels allow for fast water transport as the slip length [158,159] of the flow exceeds the width of the channel [160]. Electrostatic interactions between the surface charge of the nanochannel and mobile ions can lead to a “unipolar solution” inside the nanoconfined space as the Debye length is comparable to the radius of the channel [161,162,163]. It has been also demonstrated that dielectric mismatch between the electrolyte and the pore surface has profound influence on the ion transport through the nanopore [164]. In principle, interesting transport phenomena are expected when the dimension of the pore approaches to the characteristic lengths of molecular interactions. The effect of nanoconfinement is even stronger upon coating the nanopore with polymers of similar size to the pore dimension as the end-tethered polymers pose additional constraints on the physical interactions and chemical reactions in the system. The nanopore can further acquire gating properties when the coating polymers are responsive to external stimuli [165,166]. In this section, we will briefly review the theoretical descriptions of polymer-coated nanopores with a focus on their molecular structure and stimuli responses.

Homopolymers comprised of identical monomers are the most common polymeric materials for nanopore functionalization. Homopolymers with an end-to-end length on the same order of magnitude or larger than the characteristic lengths of the nanopore (i.e., the pore diameter and length) are expected to be strong gating agencies of the system. The morphology of homopolymer has been theoretically shown to be sensitive to its grafting position of the pore [167]. Changing the grafting point from the middle of inner pore surface to the pore exits, and further to the outer surface of the pore can change the orientation and spatial distribution of the free-end of the polymer. For nanopores and short nanochannels, such a boundary effect can have profound impact on the transport of molecular cargoes through the pore. For example, a vestibular pooling mechanism has been predicted to dominate the cross-pore translocation of large cargoes that are affinitive to the coating polymer of the pore [168]. For long nanochannels, the boundary effect is often neglected for theoretical simplification so that the polymer profile can be solved with periodic boundary conditions. By assuming the rotational symmetry of the polymer layer, the mathematical problem of the nanochannel can be further reduced to one-dimensional, which is convenient for constructing mean field and scaling theories. Scaling arguments predicted qualitative phase diagrams for neutral homopolymers grafted to the inner surface of nanochannel, as a function of the channel width and the grafting density of the polymers [169]. Such analyses have been applied to both the good and poor solvent conditions, which leads to the prediction that under certain coating conditions the nanopore can be switched between open and closed states by changing the solvent quality [170]. Guided by these predictions, MD simulations have been carried out to study the effect of co-nonsolvent on homopolymer-coated nanopores [170]. As shown in Figure 9A, co-nonsolvent facilitates the passage of nanoparticles by collapsing the coating polymers. Figure 9B shows the morphology switching process of neutral homopolymer that is triggered by co-nonsolvent. A similar switching process can happen inside a nanopore coated by polyelectrolytes, with the trigger being multivalent counterions such as calcium that can bridge two negatively charged polymer segments (Figure 9C). The conductance of the nanopore has been calculated by a molecular theory [151] as a function of the calcium concentration and the pH (Figure 9D). The theoretical results reveal that the ionic current inside the nanopore can be controlled not only by the calcium concentration but also by the pH.

A long-standing challenge it is to find a rigorous theoretical solution of the mass transport profile inside a nanopore, especially when the pore is coated with polymers. For simplicity, many theoretical studies assume the coating polymer brush to have uniform density distribution perpendicular to the surface, or fixed height that is not dependent on environmental conditions. Efforts have been made to go beyond such crucial simplifications by optimizing the free energy of the system as a functional of the polymer density field. Das group has developed a Strong Stretching Theory (SST) in which the stretching of the end-grafted polymer is taken into account [171]. In an augmented version of the SST, the effect of excluded volume among the polymers is considered [172]. The model has been applied to study the electrokinetic energy conversion in polyelectrolyte-coated nanochannels and showed that electroosmotic transport in such a system can be stronger than expected [173]. Especially, the model predicts that the polyelectrolytes can move the electric double layer away from the grafting surface towards the center of the pore, which greatly enhances the efficiency of energy conversion. The model also demonstrates that pH can have a strong effect on the stretching of weak polyelectrolytes and therefore the transport properties inside the nanochannel. At this point, it should be stressed that the acid–base equilibrium of weak polyelectrolytes, when immobilized to surfaces, can behave very differently from those of mobile ones in bulk solution [26]. This phenomenon has been studied by different theoretical methods including scaling theory [51,174], self-consistent field approaches [50,73,175], MC simulation [70], and systematically investigated by molecular theory for different surface curvatures [18,26,27]. The shift of acid–base equilibrium can be even more significant, when the end-tethered polyelectrolytes are confined in nanopores. Moreover, such a shift of chemical reactions is not uniform within the nanopore but dependent on the location of the chemicals. Therefore, the reactions can be thought to have varying local reaction constants throughout the confined space. Nanopores coated with polyelectrolytes can serve as ionic gates whose conductivity can be turned on and off by the proton concentration in surrounding environment. One example is a long nanopore coated with poly(4-vinyl pyridine) (P4VP) brushes, which has been constructed by Yameen et al. in [176]. Molecular theory study [27] shows that the acid–base equilibrium of P4VP is shifted with the apparent $pK_{a}$ being 3.7, which is significantly lower than the bulk value of 5.2. This shifted reaction equilibrium is predicted to cause a different pH dependence of conductivity than what is expected based on the bulk $pK_{a}$ value. The theoretical prediction from molecular theory is in good agreement with the experimental observation of pH-dependent conductivity [27]. Notice that the predicted shift in the apparent $pK_{a}$ for the P4VP nanochannel is caused by the same physical mechanism as discussed previously in Section 4.1 for PE brushes and acid-ligated gold NPs. The only difference is that P4VP is a polybase while poly(acrylic acid) surface-modified NPs carry acids, thus the pH response of polybase as function of pH is oppositely the apparent $pK_{a}$ is lower instead of higher as observe for acid ligated gold NPs and PAA brushes.

The ideal solution approximation of chemical reactions in general does not apply in nanopores. Besides acid–base equilibrium, another example is ligand–receptor binding under nanoconfinement [155]. In this case, the binding curve has been predicted to span more than 10 orders of magnitude in bulk protein concentration as shown in Figure 10C. The width of the transition regime is much larger than those that are predicted by the commonly used Langmuir isotherm, regardless of the choice of dissociation constant [155]. This unexpected behavior suggests that the concept of dissociation constant does not necessarily hold in nanoconfined environment as the apparent constant itself would depend on the protein concentration as shown in Figure 10D. Similar to the acid–base chemical equilibrium in a nanoconfined environment, the equilibrium of ligand–receptor binding is shifted due to the nanoconfinement. The (effective) ligand–receptor dissociation constant has become position dependent, similar to the acid–base dissociation of Equation (Equation 6). Analogous to the acid–base chemical equilibrium constant, the apparent or effective binding constant is different from the bulk binding constant and coupled with the local environment, which is now the protein concentration, instead of proton concentration for acid–base equilibrium, as is shown in Figure 10D. As proteins bound to the coating ligands can change the ionic conductivity of the nanopore, understanding such nanoconfined binding is useful for the design of nanopore-based biosensors.

Nanopores coated with stimuli-responsive layers are promising elements in nanofluidic circuits. For the integration of multiple stimuli responses into a single nanopore, one strategy is to use copolymers made of distinct monomers. This approach is becoming more practical as the copolymer synthesis technology advances. Surface-attached copolymer brushes can be achieved by using either “grafting from” or “grafting to” methods [177]. It is even possible to control the chemical sequence of the target copolymer at the resolution of single monomer [178,179,180]. Ananth et al. has recently synthesized artificial unfolded proteins with designed amino acid sequences and attached them to the surface of a solid-state nanopore [181]. A DNA origami scaffold has been developed to further enable position control of protein grafting within nanoconfinement [182,183]. With these fast-growing experimental tools, sequence-designed polymer coatings could become a new frontier of artificial nanopores.

Theoretical designs of copolymer-coated nanopores have been attempted in the last decade. Cheng and Cao reported a MD simulation of triblock copolymer with the design of one hydrophilic charged block being flanked by two hydrophobic neutral blocks [184]. Such copolymer design leads to a temperature-responsive gating mechanism, in which the nanopore adapts a collapse-to-the-center morphology at low temperature, and a collapse-to-the-wall morphology at high temperature. The center phase requires the charged polymer block to stretch out against an entropic penalty. This penalty is, however, too large at high temperature for the center phase to be thermodynamically stable.

A molecular theory study of a responsive copolymer gate [185] has been done for a short nanopore as schematically illustrated in Figure 11A. The sequence of the copolymer is shown in Figure 11B. The copolymer gate is designed to have two blocks with different charge and hydrophobicity. The first block is neutral and hydrophobic, while the second block contains alternating neutral and ionizable monomers (one copolymer gate employs only one kind of ionizable monomers in the hydrophilic block, either acidic or basic). The second block is attached to the wall and responsive to the proton concentration of the solution as the hydrophobicity of ionizable monomers is pH-dependent. The competition between hydrophobic attraction and electrostatic repulsion shapes the gating structure of the nanopore. When the hydrophobic attraction dominates, the copolymer will collapse to the wall of the nanopore. With enough electrostatic repulsion, the end-tethered block will stretch out to enable the neutral block to collapse into the pore center. Taking into consideration the coupling between physical interactions and chemical reactions, molecular theory predicts a phase diagram of the copolymer-coated nanopore (Figure 11C), which shows that besides pH, salt concentration can also strongly impact the gating of the copolymer. This unexpected result highlights the dual role of salts in screening the electrostatic interactions and shifting the acid–base equilibrium. The transition between wall and center phases is rather abrupt as shown in Figure 11D, which could happen in a wide range of pH (marked gray in Figure 11E) but most likely when the free energy match between the two states. The ionization fraction of the copolymer can experience a discontinuous change at the transition pH, which can be used for logic gating of ions. The above examples of theoretical nanopore design demonstrate the potential of sequence-controlled polymers as multifunctional nanofluidic gates.

### 4.6. The Nuclear Pore Complex

As the largest biological nanopore and the sole intracellular channel in eukaryotic cells, the nuclear pore complex (NPC) controls the biomass transport in and out of the cell nucleus in a highly selective yet efficient way [186]. The overall architecture of the NPC is a “hairy” pore [187] as schematically depicted in Figure 12A. Unlike most of the transmembrane protein channels whose gating relies on the conformational change of folded proteins, the NPC uses hundreds of intrinsically disordered proteins (IDPs) as its gatekeeper. These IDPs are called FG-nucleoproteins (FG-Nups) as they contain many hydrophobic phenylalanine-glycine (FG) repeats separated by hydrophilic spacers. More than 10 kinds of FG-Nups form a selective permeability barrier, which transports water, ions, and macromolecular cargoes with nuclear import/export signals (short amino acid sequences) while blocking unrecognized macromolecules. Karyopherins (Kaps) that can bind to the FG repeats through hydrophobic interactions serve as transport agents for large cargoes and shuttle between the nucleus and cytoplasm. Given the unfolded and flexible nature of the FG-Nups, the NPC is essentially a copolymer-coated nanopore with the sequences of the copolymers optimized by biological evolution. Therefore, structural knowledge of the NPC is not only of fundamental importance in biology, but also valuable in guiding the rational design of copolymer-coated artificial nanochannels. However, despite the experimental progress in understanding the scaffold structure of the NPC [188,189], the functional structure of the permeability barrier still remains elusive and debated by different hypotheses [190,191,192,193]. The small size of the pore, fast dynamics of the FG-Nups, together with the aqueous working environment pose a great challenge for the experimental resolution of the molecular gate. Modeling and simulation efforts are especially crucial to elucidate the gating structure and reconcile conflicting views [194].

Various theoretical methodologies have been applied to study the yeast NPC, using different levels of coarse-graining to deal with the molecular complexity of the system [195,196,197,198,199,200,201]. In classical mean field treatments, the FG-Nups are modeled as homopolymers with each monomer representing one or several amino acids. The electrostatic, van der Waals, and hydrophobic interactions between amino acids are absorbed into a Flory–Huggins type χ parameter that describes the overall cohesiveness between the coarse-grained monomers. With such simplification, mean field theory provides an efficient tool to investigate the structural behavior of FG-Nups under different in vitro conditions. For example, Flory-like mean field theory has been applied to describe the phase separation of FG-Nups in solution [202]. The SCF approach has been applied to study how Kaps affect the structure of FG-Nup brushes attached to surfaces of varying curvatures [203,204].

MD simulations of the FG-Nups fall into two categories: all-atom and coarse-grained. Considering the size of the NPC, all-atom simulations are so far limited to single or several FG-Nups, often of the same type such as Nsp1. Even with the full atomistic details, results from different simulations are not necessarily consistent with each other due to the differences in the chosen force fields, parameterizations, and the treatment of water. Nevertheless, all-atom simulations have provided rich molecular insights on the conformations of FG-Nups and their binding with the Kaps. Gamini et al. performed all-atom MD simulation with implicit water to predict that FG-Nups form a network of protein bundles [205]. The bundles typically consist of 2–6 proteins and are interlinked by single FG-Nups. On the other hand, all-atom simulations with explicit water suggest that FG-Nups are highly flexible and unstructured, even in the presence of multivalent Kaps [206]. Nevertheless, the physical extension of FG-Nups is found to be sensitive to the choice of force field for the explicit water [207]. Coarse-grained simulations have been conducted to study either a subset of FG-Nups under nanoconfinement or the whole NPC. By comparing to reference systems integrated with mutated FG-Nups, coarse-grained simulations results highlight the importance of charge and sequence effects in the structuring of the permeability barrier [198,208]. The whole NPC simulation by Ghavami et al. showed that the native FG-Nups assume a donut-like morphology [200], which is consistent with the peripheral passageway of Impβ observed in single-molecule florescence experiment, but cannot explain the central transporter revealed by electron microscopy [209].

Recent experiments showed that certain hydrophilic spacers are more cohesive than others [210], suggesting the molecular interactions are more complicated than previously thought. These cohesive spacers are distributed in sub-domains on the sequences of FG-Nups, as shown in Figure 12B. Recent experiments have also updated the anchor (Figure 12C) and stoichiometry information [189,211] of the FG-Nups, which are key players in determining the gating structure of the biological nanopore. Taking these new experimental data into consideration, a structural model of NPC has been developed within the framework of molecular theory [201]. The model predicts a mosaic distribution of different FG groups in space as shown in Figure 12D, suggesting different Kaps can take distinct transport pathways. The overall structure of the permeability barrier is a composite of condensates and low-density zones orchestrated by FG-Nups of different cohesiveness (Figure 12E). Last but not least, Figure 12F shows that the self-built electric potential throughout the NPC is highly polarized, although most of the opposite charges have similar spatial distributions with the net charge being relatively homogeneous in space. Together, these theoretical insights depict a gating picture in which the fast yet selective transport through the NPC is enabled by multiple routing mechanisms including FG, entropic, and electrostatic steering. The sequence–structure–function relation of the FG-Nups can be a great source of bioinspiration for the rational design of programmable nanopores functionalized by sequence-controlled copolymers.

## 5. Concluding Remarks

In the previous sections, we presented an overview of some of the main theoretical methods that we consider key in the field of weak PE at interfaces, as well as salient applications of them in the study of engineered NPs and nanopores. We focused mainly on the acid–base equilibrium of end-tethered weak PE, but also provided the reader with a couple of examples of other chemical equilibria that have been described using theoretical methods in recent years.

End-tethered weak polyelectrolytes have been theoretically studied by a variety of methods. Ranging in level of details we can mention scaling theory, analytical and numerical SCF theory, SCF-SCF approach, MT, and molecular simulations (both MC and MD). They essentially differ on the level of coarse-graining and molecular detail in which the polyelectrolyte chain are represented. We particularly focused on mean-field theories and only briefly mentioned molecular simulations, as most simulations of weak PE so far are focused on weak PE in solution and not on weak polyelectrolyte near interfaces or confined dense spaces; the subject on this paper. We like to mention that there exist several other theoretical approaches to describe tethered polymers such as classical density-functional theory and PRISM-like theories [212,213]. However, as these theories have only been applied to strong polyelectrolytes so far [141,214], we did not discuss them here.

A common feature that emerges from the studies of weak PE near interfaces is that their acid–base chemical equilibrium and chemical state is very different from that of the same acid or base monomer in solution. Throughout the different examples of nanosystems presented in Section 4, it is found that there is a strong coupling between the local physical forces and chemical reactions. There is a delicate interplay between various physical interactions, such as steric repulsion, confinement, electrostatic interactions, hydrophobic interactions, and chemical interactions such as acid–base and LR binding, which can lead to counterintuitive and non-trivial results. Nanopores, nanoparticles, and planar brushes all behave very differently because of the curvature. In summary, the structural organization of the system is strongly coupled with its chemical state. This coupling has profound consequences on, for example, the stability of PE-coated NPs, the amount of adsorbed proteins on weak PE brush, the conformational state of biomacromolecules, as well as the ability of synthetic and biological nanopores to conduct ions or translocate cargoes. The take-home message is that the chemical equilibrium state of PEs confined in an interface can be vastly different from their chemical state in dilute solution. Theoretical modeling can be used to understand both synthetic and biological systems that consist of confined weak polyelectrolyte, paving the road to providing guidelines for the rational design of smart responsive materials and nanosystems.

The study of weak polyelectrolytes that are confined in an interface is challenging but also a very interesting field. There are a vast number of applications for potential future investigations. One could expect more simulations that explicitly address charge regulation of dense interfacial polymeric systems. Another interesting avenue of research involves the investigation of the properties of confined polyampholytes, meaning systems containing both polyacids as well as polybases or copolymers containing both acid and base monomers. In a larger sense, many biological systems, such as proteins and the nuclear pore complex, are polyampholytes because they consist of amino acids. Another biological relevant and interesting system is for instant chromatin, the complex of DNA and proteins found in the nucleus of eukaryotic cells. In theoretical studies of systems containing polyampholytes, one needs to carefully consider the validity of the electrostatic mean-field approximation. This approximation can break down, as positive and negative charges can result in a mean-field electrostatics potential that is effectively zero, given that short range electrostatic correlation are not properly considered.

We like to point out that theories and results discussed in this review that rely on mean-field approximations do not include short-range electrostatic correlations. These correlations can become important and can cause strongly charged polyelectrolytes in good solvent conditions to collapse if the Bjerrum length, $l_{b}$, is comparable or greater than the distance between the charges, *b*, i.e., $l_{b}/b≳1$ [215]. As weak polyelectrolytes reduce the amount of charged monomers via charge regulation, this effectively increases the distance the charges, making it easier to satisfy above inequality. Therefore, only for full chargeable polyelectrolytes, where *b* can be comparable to $l_{b}$ short-range correlations can become important. For intermediate pH values and sufficient separation between chargeable sites the electrostatic mean-field approximation is valid, as corroborated by the agreement between mean-field MT predictions and experimental observations for several different curved and flat nanosystems, including polymer functionalized nanochannels [27], nanoparticles [26], and planar surfaces [24,28].

Short-range electrostatic correlations are also important for polyelectrolytes in the presence of multivalent ions, as they can enhance the strength of electrostatic interactions [216]. This can result in the formation of ion-bridges [30]. Likewise for polyampholytes, which contain both positive and negative charges, the electrostatic mean-field approximation can break down. Theoretical description of short-range electrostatic correlations beyond mean-field have been developed for strong polyelectrolytes in solution and melts. See, for example, in [212,217,218,219,220,221]. However, so far, none of these approaches have been extended or applied to tethered weak polyelectrolytes. Nonetheless, effects such as ion-pairing, ion-condensation, and ion-bridging can be described on a mean-field level by considering these physical processes as chemical reactions (as mentioned in the result section).

## Figures and Tables

**Figure 1 polymers-12-02282-f001:**
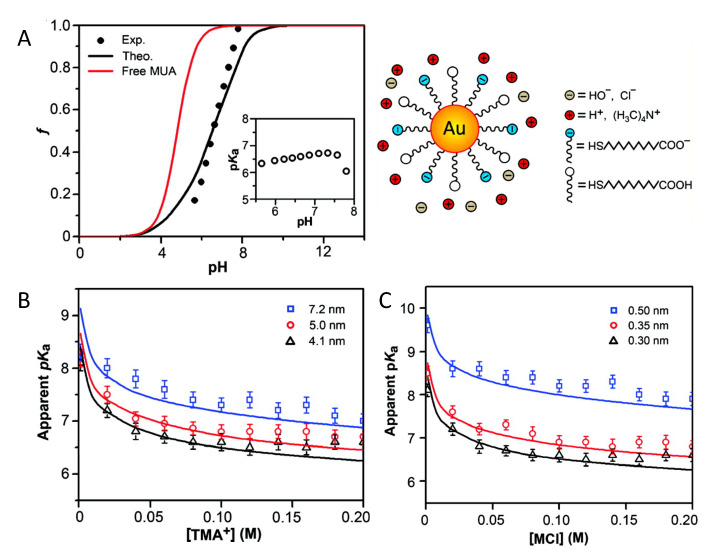
(**A**) Fractions of dissociated mercaptoundecanoic acid (MUA) ligands as a function of pH for a NP of D = 4.1 nm. The theoretical charge fraction (black curve), *f*, was calculated with the molecular theory (MT). The calculated value corresponding to free MUA in solution using the Henderson–Hasselbalch equation is plotted in red. The inset shows the experimental $pK_{a}$ of NP’s ligands. The salt concentration (tetramethylammonium chloride (TMACl)) is 0.08 M. (**B**) The apparent $pK_{a}$ of Au-MUA NPs of different sizes plotted as a function of TMACl concentration. (**C**) The apparent $pK_{a}$ of Au-MUA NPs of D = 4.1 nm as a function of the salt concentration for different salt cations (the anion is always Cl${}^{-}$). R = 0.30, 0.35, and 0.50 nm correspond to the radii of TMA${}^{+}$, TEA${}^{+}$, and TBA${}^{+}$, respectively. Open markers correspond to experimental data; lines were calculated with MT. $pK_{a}$ for free MUA is ~4.8. (Reprinted with permission from the authors of [26]. Copyright 2011 American Chemical Society).

**Figure 2 polymers-12-02282-f002:**
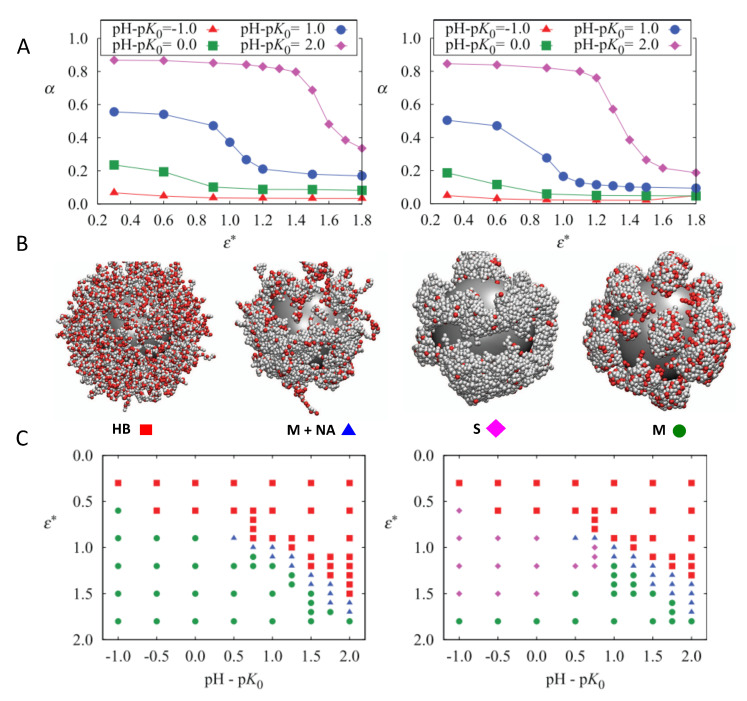
(**A**) Ionization fraction as a function of attraction strength for different values of pH − pK${}_{0}$. Upper and lower figures correspond to grafting densities σ = 0.2 nm${}^{-2}$ and σ = 0.6 nm${}^{-2}$, respectively. (**B**) Examples of the four observed polymer morphologies (from left to right): homogeneous brush, micelles plus non-aggregating chains, stripes, and micelles. Charged (unprotonated) monomers are shown in red and uncharged in light gray. Salt ions are omitted. (**C**) Morphology diagrams as a function of attraction strength and pH − pK${}_{0}$ for a nanoparticle (NP) grafted with weak polyacid chains and σ = 0.2 nm${}^{-2}$ (left) and σ = 0.6 nm${}^{-2}$ (right). Red squares indicate a homogeneous brush, green circles indicate micelles, blue triangles indicate micelles plus non-aggregating chains, and purple diamonds indicate stripes. (Reprinted with permission from the authors of [70]. Copyright 2012 American Institute of Physics).

**Figure 3 polymers-12-02282-f003:**
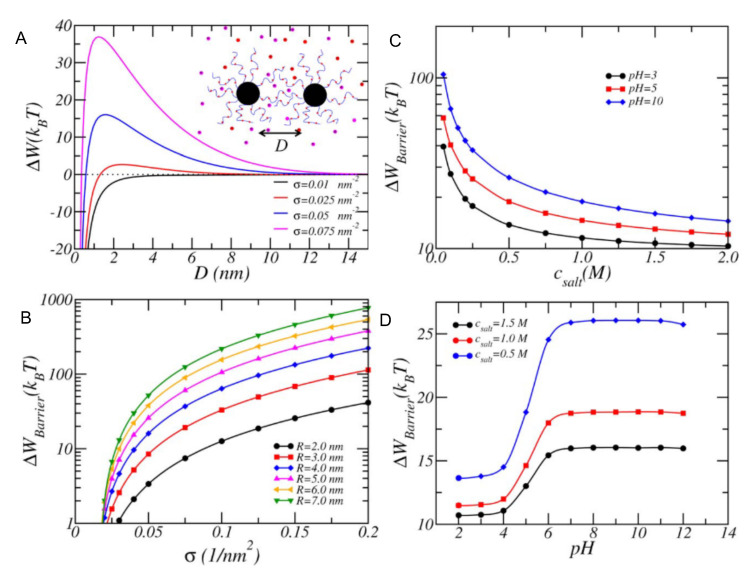
Stability of NPs coated with pAA-a-AMPS polyelectrolytes with *N* = 50 monomers (ratio AA:AMPS = 1:1). The dissociation constants are K${}_{ion}$(NaCl) = 0.6 M${}^{-1}$ (ion-pairing), K${}_{dis}$ (Na-AMPS and Na-AA) = 8.07 M (ion condensation), and $pK_{a}$ = 5.0 (deprotonation). NP concentration = 10 nM (**A**) Free energy versus separation between NPs with different pAA-a-AMPS surface coverages, as indicated in the legend. NP radius is R = 4 nm. Brine-like conditions: pH = 10 and salt concentration of NaCl = 1.5 M. (**B**) Free energy barrier height or maximum of free energy as a function of surface coverage for different NP core radii. Other conditions as in panel A. (**C**) Barrier height versus salt concentration for various pH values. Conditions: NP radius R = 4 nm, polymer surface coverage σ = 0.05 nm${}^{-2}$. (**D**) Barrier height versus pH for various salt concentrations. Conditions as in panel C. (Reprinted with permission from the authors of [93]. Copyright 2014 Wiley Periodicals, Inc.).

**Figure 4 polymers-12-02282-f004:**
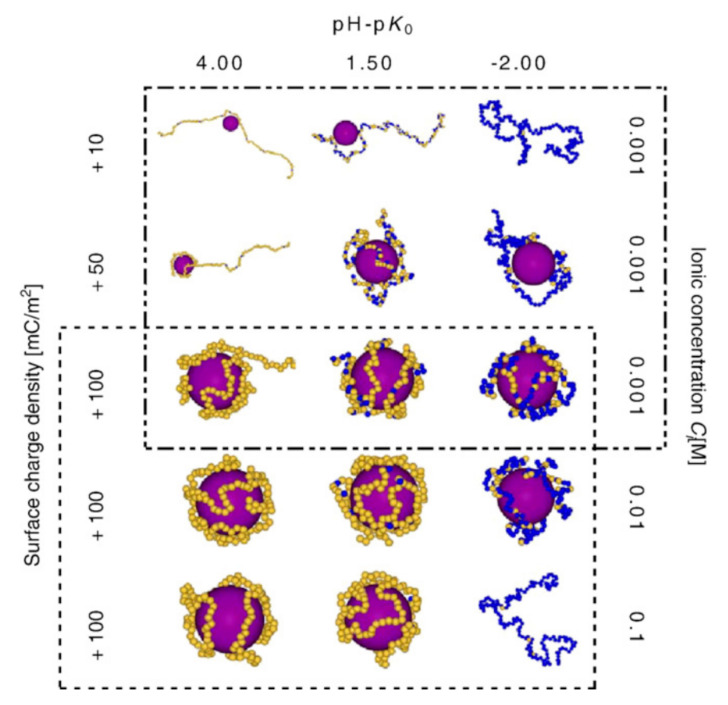
Conformations of complexes of a weak and flexible acid PE with *N* = 160 monomers and an oppositely charged nanoparticle with radius R = 3.57 nm, for different pH and ionic strength of the bulk solution and surface charge density, as indicated in the plot. Yellow spheres represent charged deprotonated momoners; blue spheres represent uncharged protonated ones. (Reprinted with permission from the authors of [99]. Copyright 2006 Elsevier).

**Figure 5 polymers-12-02282-f005:**
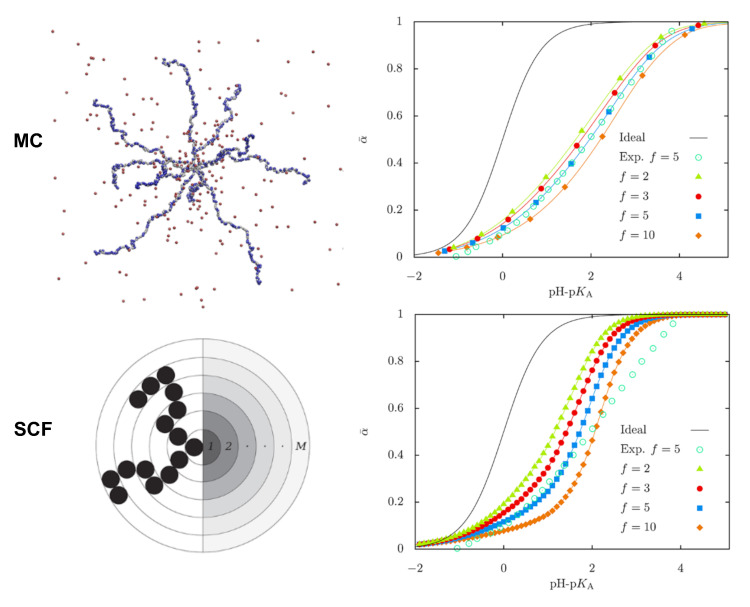
(Right panel) Representative MC conformation of a weak polyacid star containing 10 arms (**top left**) and illustration of the corresponding layer structure in the Scheutjens–Fleer Self-consistent (SF-SCF) model (**bottom left**). (Left panel) Average degree of dissociation as a function the difference pH −$pK_{a}$, corresponding to MC simulations (**top right**) and SF-SCF calculations (**bottom right**) for stars of different number of arms (*f*), as indicated in the legend. The ideal titration curve corresponds to that of an isolated segment. (Reprinted with permission from the authors of [109]. Copyright 2014 American Chemical Society).

**Figure 6 polymers-12-02282-f006:**
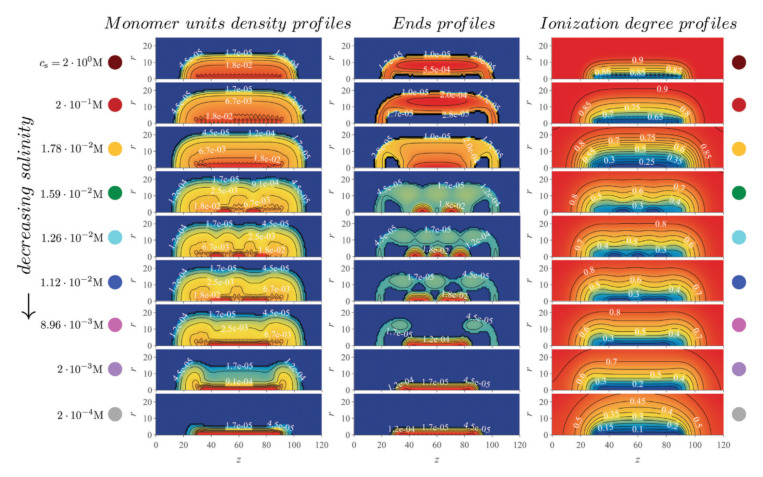
Molecular brush consisting of a semi-rigid polyelectrolyte (PE) backbone with *N* = 61 monomers and 21 flexible side chains with *N* = 30 monomers immersed in a aqueous solution. Monomers are weak polyacids. Panels correspond to the spacial distribution of the monomers (left), that of the terminal groups (middle) and the ionization degree (right), at different salt concentrations, as indicated in the graph. pH − pK${}_{a}$ = 1 and solvent quality parameter χ = 1.5. The distributions show changes in the density from denser (red color) to less dense (blue color). (Reprinted with permission from the authors of [113]. Copyright 2020 The Royal Society of Chemistry).

**Figure 7 polymers-12-02282-f007:**
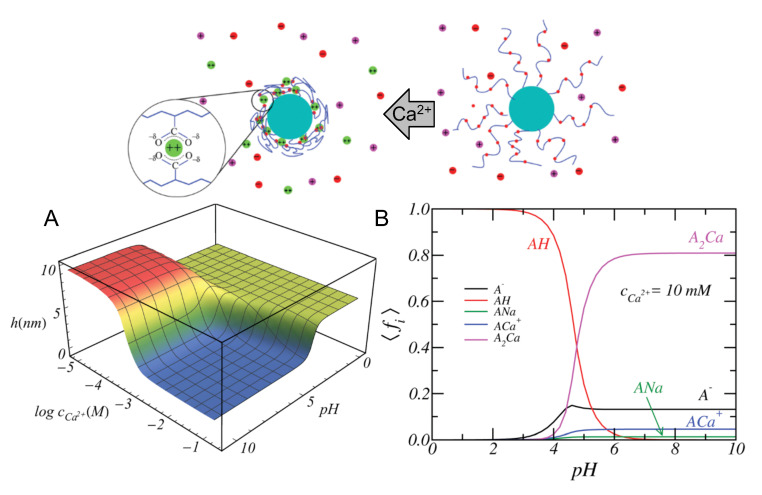
(upper panel) Illustration of the effect that calcium ions can have on the structure of a poly(acrylic acid) layer grafted to a NP. The inset shows the bridge formed between one calcium ion and two carboxylic acid monomers. (**A**) Height of the poly(acrylic acid) layer as a function of the bulk pH and Ca${}^{2+}$ concentration. (**B**) Average fraction of A${}^{-}$, AH, ANa, ACa${}^{+}$, and A${}_{2}$Ca as a function of pH for a 10 mM concentration of of Ca${}^{2+}$ ions in solutions. The NaCl concentration is 150 mM (physiological conditions). In all figures, the NP radius is R = 4 nm, the surface coverage is σ = 0.15 nm${}^{-2}$, and the number of monomers is *N* = 50. (Reprinted with permission from the authors of [30]. Copyright 2018 The Royal Society of Chemistry).

**Figure 8 polymers-12-02282-f008:**
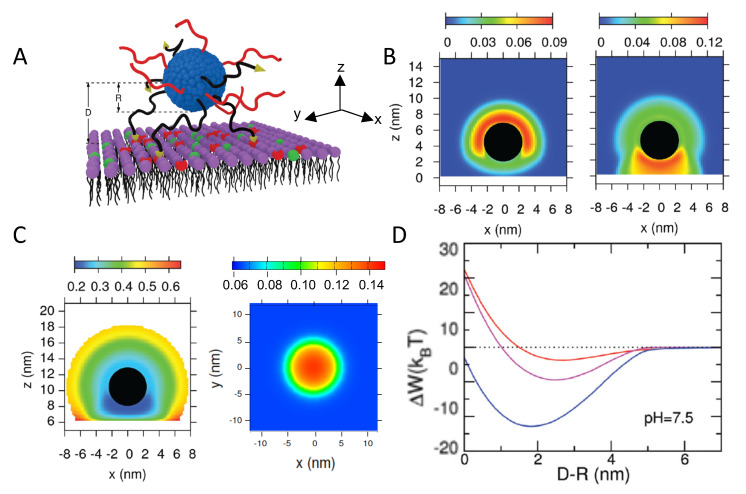
(**A**) Schematic illustration of a polymer-coated NP interacting with a lipid membrane. Neutral polymer chains are in red, while the polybases ($pK_{a}$ = 7.5) with their ligand-end group are in black. Uncharged lipids are in purple, overexpressed receptors in red, and negatively charged lipids in green. (**B**) Contour maps of the polymer volume fractions for both neutral polymers (left panel) and polybase (right panel) for a NP of radius R = 2.5 nm close to the lipid layer for a pH = 7.5 and a salt concentration of 0.10 M. Surface density σ = 0.20 nm${}^{-2}$, *N* = 20 segments. (**C**) Coated NP interacting with a lipid membrane with both overexpressed receptors and charged lipids. Left panel: Contour map of the fraction of charged groups of the polybases on a NP close to the lipid membrane. The conditions are the same as in panel (b). Right panel: Contour map of the fraction of charged lipids in the membrane. The center of the NP is at (x; y) = (0; 0) and 2.0 nm above the membrane surface. The conditions are the same as in panel (b). (**D**) Free energy as a function of the distance between the NP surface and the lipid layer. The conditions are the same as in panel (b). The colors correspond to the three membranes modeled: no overexpressed receptors and negatively charged lipids (red), neutral lipid membrane with overexpressed receptors (magenta), and membranes with both overexpressed receptors and charged lipids (blue). (Reprinted with permission from the authors of [156]. Copyright 2013 The Royal Society of Chemistry).

**Figure 9 polymers-12-02282-f009:**
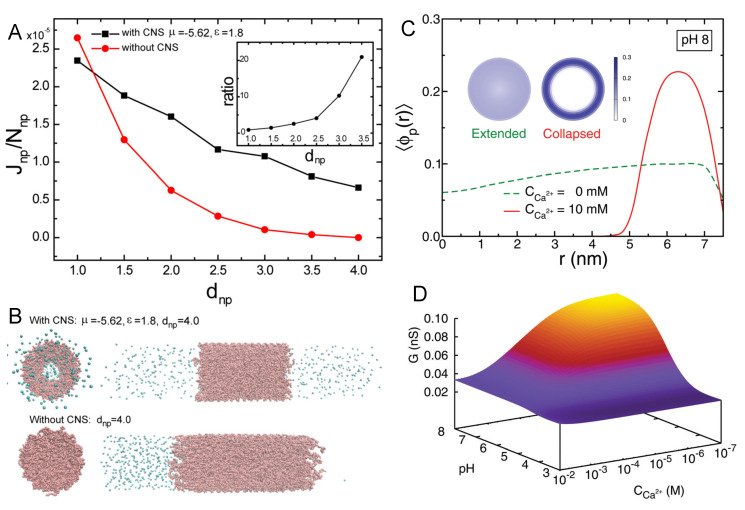
Gating of homopolymer-coated nanopores. (**A**) Stationary fluxes of nanoparticles as a function of the diameter of the NP in open (black squares) and closed (red circles) states controlled by the co-nonsolvent. The ratio of the fluxes is shown in the inset. (**B**) Morphologies of open (upper) and closed (lower) states in MD simulations, with red and green beads representing monomers and NPs, respectively. Reprinted with permission from the authors of [170]. (**C**) Profile of polymer volume fraction as a function of the distance *r* from the center of the channel, in extended (green) and collapsed (red) states. (**D**) A typical landscape of conductance as a function of pH and calcium concentration. (Reprinted with permission from the authors of [151]).

**Figure 10 polymers-12-02282-f010:**
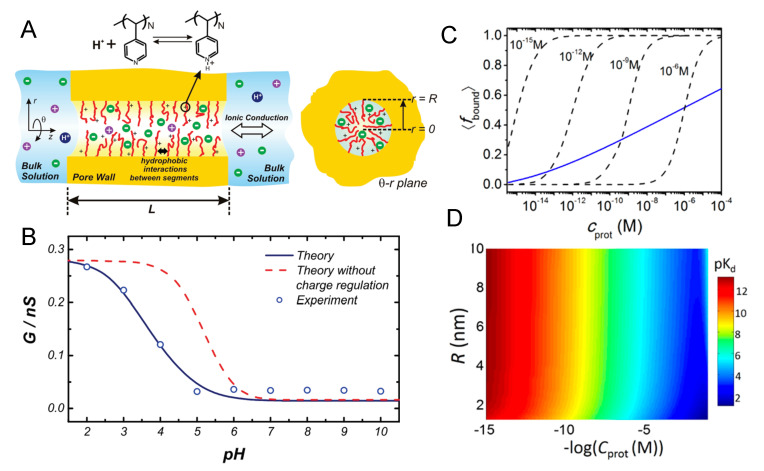
Shifted equilibria of chemical reactions in polymer-coated nanopores. (**A**) Schematic representation of the polyelectrolyte brush modified long nanopore. The inner surface of the pore is modified with end-tethered chains of poly(4-vinyl pyridine). (**B**) Comparison between experiment and molecular theory on the pH-dependent conductivity. Reprinted with permission from the authors of [27]. (**C**) Molecular theory prediction of average fraction of bound ligands inside a nanopore as a function of the molar concentration of the proteins in bulk solution (solid blue line), in comparison with the predictions of the Langmuir isotherm for different values of the dissociation constant (black dashed lines). (**D**) Apparent dissociation constant as a function of the pore radius and the molar concentration of the proteins in bulk solution. (Reprinted with permission from the authors of [155]).

**Figure 11 polymers-12-02282-f011:**
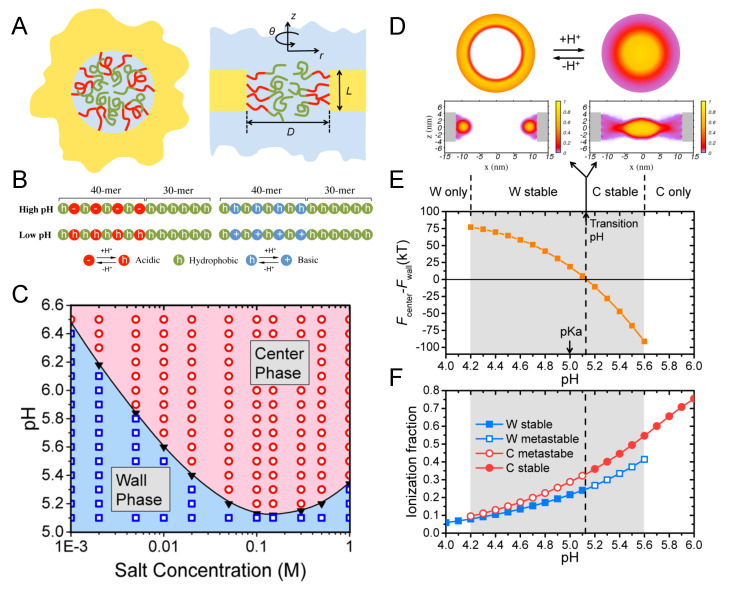
Design of copolymer for smart (stimuli-responsive) nanopore. (**A**) Schematic representation of the copolymer grafted short nanopore. (**B**) Sequence design of the copolymer. The copolymer contains in total 70 monomers, which can be classified into two groups according to their ionizability. The ionizable ones are hydrophilic when charged and hydrophobic when they are neutral. (**C**) Phase diagram of the nanopore. (**D**) Switching between center and wall phases at the transition pH. (**E**) Free energy difference between the two phases and the metastable regime. (**F**) Ionization fraction of both phases within the metastable regime. (Reprinted with permission from the authors of [185]).

**Figure 12 polymers-12-02282-f012:**
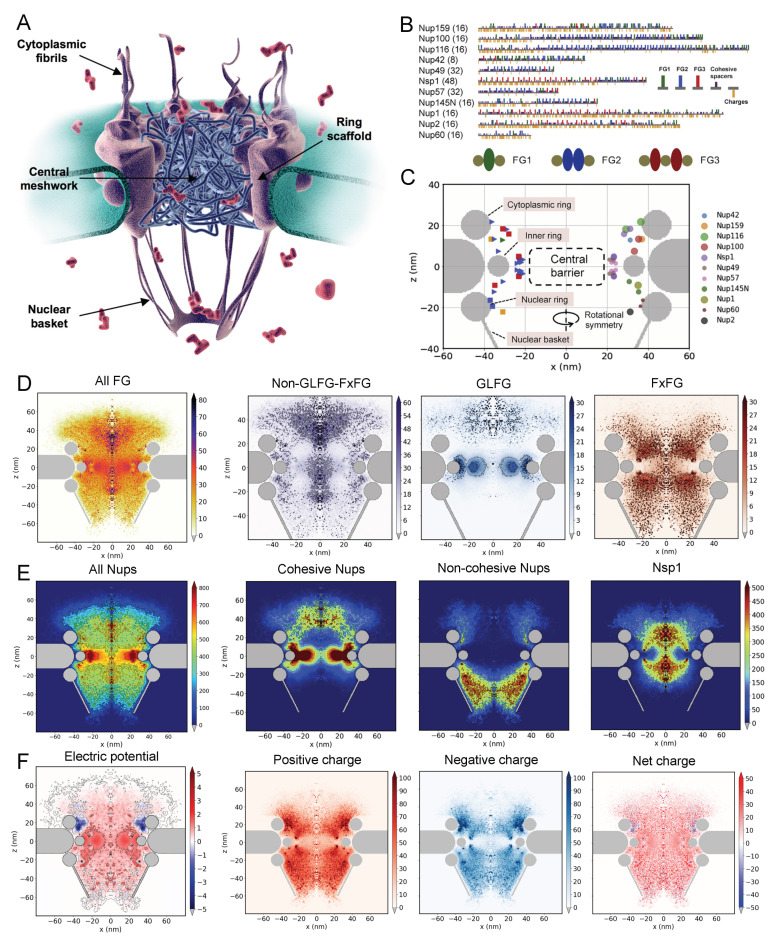
Molecular theory of the yeast NPC. (**A**) Schematic representation of the NPC. Reprinted with permission from the authors of [187]. (**B**) Stoichiometry and amino acid sequences of the FG-Nups. (**C**) Coarse-grained scaffold structure and anchor positions of the FG-Nups. (**D**) Spatial distributions of FG groups. (**E**) Spatial distribution of FG-Nups. (**F**) Electric potential and charge distribution throughout the NPC. All the color maps for amino acid concentrations are in the units of mM. The color map for electric potential is in the units of mV. (Reprinted with permission from the authors of [201].)

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
