# Peer review of "Theoretical Modeling of Chemical Equilibrium in Weak Polyelectrolyte Layers on Curved Nanosystems"

_polymers, 2020, doi:10.3390/polym12102282_

Round 1

Reviewer 1 Report

As the authors of the manuscript under review state, their purpose is “to present an overview of some of the theoretical methods they consider key in the field of weak polyelectrolytes (PE) at interfaces, to enable a better understanding of the vrarious theoretical approaches”. In general, this task seems to be done quite well but one important, as to my opinion, exception. Namely, the authors fully ignore the fact, that the charge density correlations can significantly contribute into conformational behavior and the total free energy of the system. In this connection one should mention the papers by JF Marko on the lateral layering of the brushes in earlier 90-th. 

I could also mention the papers by Borue and Erukhimovich (1988, 1990) and their recent followers who estimated quantitatively the correlation contribution into the total free energy of PE as compared with that of Debye-Huckel. Ignorance of these methods is partially justified, though. since they were not applied to surface systems.  

I also found few misprints.

Summarizing, I recommend publishing the review in MDPI.

Author Response

Reviewer 1:

Reviewer’s Comment #1: As the authors of the manuscript under review state, their purpose is “to present an overview of some of the theoretical methods they consider key in the field of weak polyelectrolytes (PE) at interfaces, to enable a better understanding of the various theoretical approaches”. In general, this task seems to be done quite well but one important, as to my opinion, exception. Namely, the authors fully ignore the fact that the charge density correlations can significantly contribute to conformational behavior and the total free energy of the system. In this connection one should mention the papers by JF Marko on the lateral layering of the brushes in earlier 90-th.

Authors’ response: The reviewer addresses an interesting issue regarding the limitations of the mean-field approximation to deal with correlations in charge fluctuations. The paper by J.F. Marko and Y. Rabin (Microphase separation of charged diblock copolymers: melts and solutions, J. F. Marko and Y. Rabin, Macromolecules 1992, 25, 5, 1503-1509.) is indeed very suited in that direction, introducing correlations for strong polyelectrolytes melts and solutions. We have added the reference along two paragraphs in the Concluding Remarks section addressing this question (the addition is marked in blue in the manuscript). The new text reads:

“We like to point out that theories and results discussed in this review that rely on mean-field approximations do not include short-range electrostatic correlations. These correlations can become important and can cause strongly charged polyelectrolytes in good solvent conditions to collapse if the Bjerrum length, lb, is comparable or greater than the distance between charges, b, i.e., lb / b  1. [215] Since weak polyelectrolytes reduce the amount of charged monomers via charge regulation, this effectively increases the distance between the charges, making it easier to satisfy above inequality. Hence only for full chargeable polyelectrolytes, where b can be comparable to lb short-range correlations can become important. For intermediate pH values and sufficient separation between chargeable sites the electrostatic mean-field approximation is valid, as corroborated by the agreement between mean-field MT predictions and experimental observations for several different curved and flat nanosystems, including polymer functionalized nanochannels [27], nanoparticles [26], and planar surfaces. [24,28].

Short-range electrostatic correlations are also important for polyelectrolytes in the presence of multivalent ions, since they can enhance the strength of electrostatic interactions.[216] This can result in the formation of ion-bridges. [30] Likewise for polyampholytes, which contain both positive and negative charges, the electrostatic mean-field approximation can break down. Theoretical description of short-range electrostatic correlations beyond mean-field have been developed for strong polyelectrolytes in solution and melts. See for example Refs. [212, 217–221]. However, so far, none of these approaches have been extended or applied to tethered weak polyelectrolytes. Nonetheless, effects such as ion-pairing, ion-condensation, and ion-bridging can be described on a mean-field level by considering these physical processes as chemical reactions-as mentioned in the result section.”

Beyond the reviewer’s suggestions, we have also added the following references to support the added text: Namely references 215, 216, 220 and 221:

  • González-Mozuelos, P.; Olvera de la Cruz, M. Ion condensation in salt-free dilute polyelectrolyte solutions. J. Chem. Phys. 1995, 103, 3145-3157. doi:http://dx.doi.org/10.1063/1.470248
  • Solis, F.J.; Olvera de la Cruz, M. Attractive interactions between rodlike polyelectrolytes: polarization,crystallization, and packing.Phys. Rev. E1999,60, 4496
  • Santangelo, C.D.; Lau, A.W.C. Effects of counterion fluctuations in a polyelectrolyte brush.Eur. Phys. J. E 2004,13, 335–344. doi:10.1140/epje/i2003-10077-7.
  • Sing, C.E.; Zwanikken, J.W.; Olvera de la Cruz, M. Electrostatic control of block copolymer morphology. Nat. Mater. 2014,13, 694–698. doi:10.1038/nmat4001

Finally, we moved the last paragraph of the Conclusions of the original version of the manuscript before the added text for readability. The change is marked in red in the revised version.

Reviewer’s Comment #2: I could also mention the papers by Borue and Erukhimovich (1988, 1990) and their recent followers who estimated quantitatively the correlation contribution into the total free energy of PE as compared with that of Debye-Huckel. Ignorance of these methods is partially justified, though. since they were not applied to surface systems.  

Authors’ response: We thank the reviewer for the references recommended. Similar to the Marko and Rabin paper, they also include charge correlations, improving mean field description of the systems: 

  • A statistical theory of weakly charged polyelectrolytes: fluctuations, equation of state and microphase separation, Yu. Borue and I. Ya. Erukhimovich, Macromolecules 1988, 21, 11, 3240-3249. 
  • A statistical theory of globular polyelectrolyte complexes, Yu. Borue and I. Ya. Erukhimovich, Macromolecules 1990, 23, 15, 3625-3632.

It should be noted though that the polyelectrolytes in those papers correspond to strong polyelectrolytes, in the sense that they are permanently charged irrespective of the conditions of the solution (pH and salt concentration). The title of the first paper could be misleading, since “weakly charged polyelectrolytes” refer actually to only a few amounts of charges on the polymer chain. This departs from the focus of our review on weak PE at curved geometries, but following the reviewer’s suggestion, we have added to our references and addressed this comment in the Concluding Remarks section, as mentioned above.

Reviewer’s Comment #3: I also found a few misprints.

Authors’ response: We thank the reviewer for carefully reading the manuscript. We have gone over it to correct misprints.

Reviewer’s Comment #4: Summarizing, I recommend publishing the review in MDPI.

Authors’ response: We are very grateful for the reviewer’s comments and their recommendation on the manuscript.

Reviewer 2 Report

In this manuscript the authors provided a nice review on the theoretical modelling of weak polyelectrolyte systems. After a concise introduction of the main theoretical models and methods, the authors summarized some very interesting applications of the theory. In my view, this review article will be highly appreciated by researchers in polymer physics. 

I only have two comments for the authors to consider when they revise their manuscript.

  1. The theoretical framework is mostly within the mean-field theory. It is well-known that correlations of fluctuations are ignored in mean-field theory. It would be very helpful if the authors could provide some more discussions on this point. Especially, why does the mean-field theory work well for the applications reviewed in this manuscript? What are the problems where fluctuation effects cannot be ignored? What are the possible approaches to improve the mean-field theory?
  2. One relevant reference on the extension of polymeric self-consistent field theory to polyelectrolytes is Shi and Noolandi, Theory of inhomogeneous weakly charged polyelectrolytes, Macromol. Theory Simul. 1999, 8, 214-229. This should be included in the references.

Author Response

Reviewer 2:

Reviewer’s Comment #1: In this manuscript the authors provided a nice review on the theoretical modelling of weak polyelectrolyte systems. After a concise introduction of the main theoretical models and methods, the authors summarized some very interesting applications of the theory. In my view, this review article will be highly appreciated by researchers in polymer physics. 

Authors’ response: We are very grateful for the reviewer’s general comments and their recommendation on the manuscript.

Reviewer’s Comment #2: I only have two comments for the authors to consider when they revise their manuscript. The theoretical framework is mostly within the mean-field theory. It is well-known that correlations of fluctuations are ignored in mean-field theory. It would be very helpful if the authors could provide some more discussions on this point. Especially, why does the mean-field theory work well for the applications reviewed in this manuscript? What are the problems where fluctuation effects cannot be ignored? What are the possible approaches to improve the mean-field theory?

Authors’ response: As for the comments of reviewer 1, reviewer 2 raises interesting questions regarding the applicability and limitations of the mean-field approximation when dealing with charged systems in which correlations between charge fluctuations are likely to occur.

We have addressed this issue in the “Concluding Remarks” section by adding a brief discussion describing the applicability and limitations and potential (future) approaches to improve the mean-field approximation. The added text is marked in blue in the revised manuscript and is also provided at large in the response to question 1 of reviewer 1. Here we like to mention that it is our understanding that there are very few publications that include corrections on fluctuations for weak PE, all of the publications devoted to short-range electrostatic correlations seem to refer to solutions or melts of strong polyelectrolytes (as the reference provided by the reviewer in the following comment).

Reviewer’s Comment #3: One relevant reference on the extension of polymeric self-consistent field theory to polyelectrolytes is Shi and Noolandi, Theory of inhomogeneous weakly charged polyelectrolytes, Macromol. Theory Simul. 1999, 8, 214-229. This should be included in the references.

Authors’ response: Per the reviewer's suggestion, we have added the Shi and Noolandi paper to the references, following also the responses on previous comments. The citation was added to section 3.2 and is correspond to new citation [46].

Reviewer 3 Report

Report on polymers 931137

The authors present a comprehensive review of the state-of-the-art in theoretical modelling of conformational properties of weak (pH-sensitive) polyelectrolytes under different conditions of confinementin curved geometries, including polyelectrolyte brushes tethered to the substrates, nanopores  or colloidal nanoparticles, polyelectrolyte molecular brushes and stars. In all these systems the effects of charge regulation (termed by the authors as chemical equilibrium) are mediated by intra- and intermolecular electrostatic interactions and are strongly dependent on and, in turn, control polymer conformations.

The review is timely, reasonably well-written, contains large number of illustrations and can be useful for theoreticians and experimentalists working in the domains of polyelectrolytes and nanocolloids. Therefore, I can recommend it for publication.

The list of references includes more than 200 items. However, I feel that references are somehow random and, as a result, a number of important references are missing:

For example, pioneering papers that identified various regimes of polyelectrolyte brushes (

  1. Pincus. Colloidal Stabilization by grafted polyelectrolytes Macromolecules 1991,24, 2912-2919

O.V. Borisov, T.M. Birshtein and E.B. Zhulina. Collapse of Grafted Polyelectrolyte Layer.  J Physique II, 1991, 1 (5), 521- 526

are missing, while instead secondary ones on the same topic (refs 12 and 13) are present; 

Also it would be good to mention that analytical SCF theories of polymer brushes were developed independently by  S.Milner, T.Witten and M.Cates (refs 36,37) and in

A.M. Skvortsov, I.V. Pavlushkov, A.A. Gorbunov, Y.B. Zhulina, V.A. Priamitsyn and O.V. Borisov                       Structure of Dense Grafted Polymer Monolayers - Polymer Science USSR 1988, 30, 1706-1715                     

Y.B. Zhulina, V.A. Priamitsyn and O.V. Borisov   

Structure and Conformational Transitions in the Grafted Chain Layers: New Theory. Polymer Science USSR 1989, 31,205-2016

while refs 35 and 38 are only secondary ones.

Furthermore, two extensive reviews  on solution properties of branched polyelectrolytes (including detailed discussion of the effects of charge regulation) and self-assembled micelles of block copolymers comprising pH-sensitive polyelectrolyte blocks cover the topics very close to that of the present review and were published in Advances in Polymers Science 2011, v.241 by Borisov et al.

Nevertheless, I can recommend the review for publication.

Author Response

Reviewer 3:

Reviewer’s Comment #1: The authors present a comprehensive review of the state-of-the-art in theoretical modelling of conformational properties of weak (pH-sensitive) polyelectrolytes under different conditions of confinement in curved geometries, including polyelectrolyte brushes tethered to the substrates, nanopores or colloidal nanoparticles, polyelectrolyte molecular brushes and stars. In all these systems the effects of charge regulation (termed by the authors as chemical equilibrium) are mediated by intra- and intermolecular electrostatic interactions and are strongly dependent on and, in turn, control polymer conformations. The review is timely, reasonably well-written, contains a large number of illustrations and can be useful for theoreticians and experimentalists working in the domains of polyelectrolytes and nanocolloids. Therefore, I can recommend it for publication.

Authors’ response: We are very grateful for the reviewer’s general comments and their recommendation on the manuscript.

Reviewer’s Comment #2: The list of references includes more than 200 items. However, I feel that references are somehow random and, as a result, a number of important references are missing. For example, pioneering papers that identified various regimes of polyelectrolyte brushes: 

  • Colloidal Stabilization by grafted polyelectrolytes, Macromolecules 1991,24, 2912-2919.
  • V. Borisov, T.M. Birshtein and E.B. Zhulina. Collapse of Grafted Polyelectrolyte Layer.  J Physique II, 1991, 1 (5), 521- 526

are missing, while instead secondary ones on the same topic (refs 12 and 13) are present.

Authors’ response: We thank the reviewer for carefully revising the references. As mentioned in the Introduction, the overview presented is focused on theoretical studies of weak polyelectrolytes (PE) at curved interfaces. Although we mention that the theory methods are derived from previous work on strong PE, a review of all the extensive work done on strong PE goes beyond the scope of this review. That is why we draw upon existing reviews on the topic, such as references 12 and 13 on scaling theories for grafted strong PE,  instead of providing an overview of the original papers. However, in light of the reviewer’s suggestion, we have added references in the methodology. The citations are [62] and [67].

Reviewer’s Comment #3: Also it would be good to mention that analytical SCF theories of polymer brushes were developed independently by  S.Milner, T.Witten and M.Cates (refs 36, 37) and in 

  • M. Skvortsov, I.V. Pavlushkov, A.A. Gorbunov, Y.B. Zhulina, V.A. Priamitsyn and O.V. Borisov, Structure of Dense Grafted Polymer Monolayers - Polymer Science USSR 1988, 30, 1706-1715.
  • B. Zhulina, V.A. Priamitsyn and O.V. Borisov, Structure and Conformational Transitions in the Grafted Chain Layers: New Theory. Polymer Science USSR 1989, 31, 205-2016. 

while refs 35 and 38 are only secondary ones.

Authors’ response: Following the previous response, references 35 and 38 correspond to comprehensive reviews on polymer brushes investigated with theoretical methods. However, per the reviewer's suggestion, we have added the following line in the methodological section, along with the references provided (the addition is marked in blue in the revised manuscript):

“The analytical theories for weak polyelectrolyte were based on original work for neutral and strong polyelectrolytes developed independently by Milner, Witten, and Cates [37] and Borisov and coworkers. [65-67]”

Reviewer’s Comment #4: Furthermore, two extensive reviews on solution properties of branched polyelectrolytes (including detailed discussion of the effects of charge regulation) and self-assembled micelles of block copolymers comprising pH-sensitive polyelectrolyte blocks cover the topics very close to that of the present review and were published in Advances in Polymers Science 2011, v.241 by Borisov et al.

Authors’ response: We thank the reviewer for the suggested reviews.

The first one (Interpolyelectrolyte Complexes Based on Polyionic Species of Branched Topology, Pergushov D.V., Borisov O.V., Zezin A.B., Müller A.H.E. (2010). In: Müller A., Borisov O. (eds) Self Organized Nanostructures of Amphiphilic Block Copolymers I. Advances in Polymer Science, vol 241. Springer, Berlin, Heidelberg) is a very interesting review of experimental and MD results on interpolyelectrolyte complexes (IPECs) of branched strong polyelectrolyte (PE) species, including PE stars, star-like micelles in aqueous solutions, and cylindrical PE brushes. On the other hand, the second review (Self-Assembled Structures of Amphiphilic Ionic Block Copolymers: Theory, Self-Consistent Field Modeling and Experiment, Borisov O.V., Zhulina E.B., Leermakers F.A.M., Müller A.H.E. (2011). In: Müller A., Borisov O. (eds) Self Organized Nanostructures of Amphiphilic Block Copolymers I. Advances in Polymer Science, vol 241. Springer, Berlin, Heidelberg) provides a comprehensive overview of theoretical work describing the self-assembly of amphiphilic ionic/hydrophobic diblock copolymers in dilute solution, considering both strong and weak monomers. Although both publications are related to our review since they address interesting applications of PE, they do depart from our main focus on weak PE at interfaces, either by referring only to strong PE (as in the first one) or to weak PE in solution (as in the second one). However, following the reviewer suggestions, we have added these references as further examples of Scheutjens-Fleer Lattice Theory in the methodology section.

Reviewer’s Comment #5: Nevertheless, I can recommend the review for publication.

Authors’ response: We thank the reviewer’s reference suggestions and the recommendation on the manuscript.